# One Step Diffusion-based Super-Resolution with Time-Aware Distillation

## Abstract

Diffusion-based image super-resolution (SR) methods have shown promise in reconstructing high-resolution images with fine details from low-resolution counterparts. However, these approaches typically require tens or even hundreds of iterative samplings, resulting in significant latency. Recently, techniques have been devised to enhance the sampling efficiency of diffusion-based SR models via knowledge distillation. Nonetheless, when aligning the knowledge of student and teacher models, these solutions either solely rely on pixel-level loss constraints or neglect the fact that diffusion models prioritize varying levels of information at different time steps. To accomplish effective and efficient image super-resolution, we propose a time-aware diffusion distillation method, named TAD-SR. Specifically, we introduce a novel score distillation strategy to align the score functions between the outputs of the student and teacher models after minor noise perturbation. This distillation strategy eliminates the inherent bias in score distillation sampling (SDS) and enables the student models to focus more on high-frequency image details by sampling at smaller time steps. Furthermore, to mitigate performance limitations stemming from distillation, we fully leverage the knowledge in the teacher model and design a time-aware discriminator to differentiate between real and synthetic data. This discriminator effectively distinguishes the diffused distributions of real and generated images under varying levels of noise disturbance through the injection of time information. Extensive experiments on SR and blind face restoration (BFR) tasks demonstrate that the proposed method outperforms existing diffusion-based single-step techniques and achieves performance comparable to state-of-the-art diffusion models that rely on multi-step generation.

## 1 Introduction

Image super-resolution (SR), a cornerstone task in low-level vision, involves reconstructing high-resolution (HR) images with intricate details from low-resolution (LR) counterparts. Owing to the inherent ill-posed nature of this task, multiple high-resolution reconstructions are plausible for a given low-resolution input, presenting a persistent and perplexing challenge. Recently, the diffusion model (Ho et al., 2020; Song et al., 2020), a novel generative model, has garnered increasing attention for its capacity to model complex data distributions. It has gradually emerged as a successor to Generative Adversarial Networks (GANs) (Goodfellow et al., 2020) in various downstream tasks, including image editing (Meng et al., 2021; Hertz et al., 2022), image inpainting (Chung et al., 2022; Lugmayr et al., 2022) and image super-resolution (Saharia et al., 2022; Yue et al., 2024).

Specifically, existing diffusion-based image super-resolution methods can be broadly categorized into two streams: one involves feeding low-resolution images along with noise into the diffusion model and training the model from scratch (Rombach et al., 2022; Yue et al., 2024), while the other (Wang et al., 2023b; Wu et al., 2024b) adapts SR tasks by fine-tuning the pre-trained text-to-image diffusion model. While these methods have demonstrated promising results, generating images typically demands tens or even hundreds of iterative samplings, significantly impeding their practical application and further advancement.

To enhance the inference efficiency of diffusion models, various acceleration techniques have been proposed, such as the development of numerical samplers (Lu et al., 2022; Zheng et al., 2024) and the applications of knowledge distillation (Salimans & Ho, 2022; Sauer et al., 2023). However,

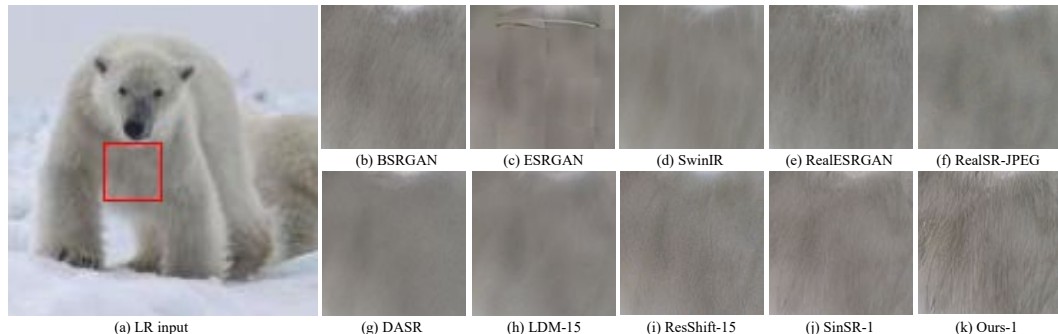

Figure 1: Qualitative comparisons on a typical real-world example of the proposed method and recent SR approaches, including BSRGAN (Zhang et al., 2021), RealESRGAN (Wang et al., 2021b), SwinIR (Liang et al., 2021), DASR (Liang et al., 2022b), RealSR-JPEG (Ji et al., 2020) LDM (Rombach et al., 2022), ResShift (Yue et al., 2024), and SinSR (Wang et al., 2023c). We mark the number of sampling steps of diffusion-based SR method with the format of "Method-n" for more intuitive visualization, where "n" is the number of sampling steps. Note that LDM contains more diffusion steps in training and is accelerated to "n" steps using DDIM (Song et al., 2020) during inference. Please zoom in for a better view.

due to the requirement of SR tasks to output images with clear details while ensuring high visual similarity with LR images, directly applying existing acceleration methods to SR tasks presents significant challenges. For the SR task, ResShift (Yue et al., 2024) has improved the sampling efficiency of diffusion-based SR models by utilizing information from LR images to reformulate the diffusion process, thereby reducing the number of sampling steps to 15. Furthermore, SinSR (Wang et al., 2023c) merges distillation techniques with a cycle consistency approach to refine the ResShift model into a single inference step. However, it only constrains the output of the student model at a single scale and fails to leverage the ability of the pre-trained diffusion model to fit diffused distributions across different time steps, a property referred to as the time-aware of the diffusion model in this paper. Recently, AddSR (Xie et al., 2024) employs adversarial diffusion distillation (ADD) (Sauer et al., 2023) for SR task to enhance sampling efficiency. Although it employs the expertise of the teacher model to optimize the student model via Score Distillation Sampling (SDS) (Poole et al., 2022), inherent biases in the gradients calculated by SDS lead to image blurring and excessive smoothness. Additionally, AddSR does not take advantage of the diffusion model's ability to extract semantic features at different levels. Instead, it relies on a pre-trained DINOV2 (Oquab et al., 2023) discriminator in pixel space, which is both expensive and challenging to optimize.

To address the aforementioned issues, we propose a time-aware distillation method that fully leverages the time-aware property of the teacher model and the latent knowledge embedded in the diffusion process. Specifically, we propose a high-frequency enhanced score distillation technique that eliminates the inherent bias in score distillation sampling and improves the high-frequency details in the student model's output by focusing on sampling in small time steps. Additionally, To overcome the performance limitations of teacher models, we incorporate adversarial learning into the distillation framework, forcing the student model to directly generate samples that lie on the manifold of real images in a single inference step. Specifically, we extract features from real and synthetic data under varying noise disturbances using the teacher model, while designing a time-aware discriminator to effectively distinguish these features. Combined with the above design, our method can match or even surpass the performance of state-of-the-art (SOTA) methods with only one-step sampling.

Overall, our contributions can be summarized as follows:

- By fully leveraging the time-aware property of the diffusion model and the latent knowledge embedded in the diffusion process, we propose a time-aware distillation method that accelerates diffusion-based SR models into a single inference step.

- We analyze the inherent bias in score distillation sampling and propose a novel score distillation method to eliminate this bias. Additionally, we focus on enhancing the high-frequency details in the student model's output by sampling at small time steps.

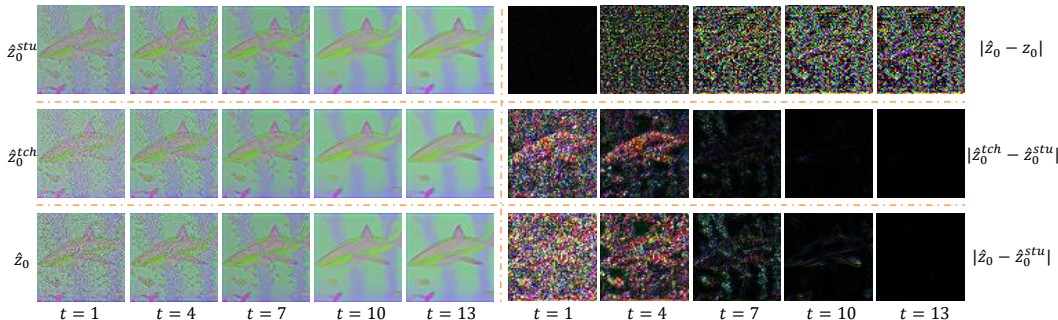

Figure 2: On the left side of the figure, we visualize the clean data predictions made by the pre-trained diffusion model after applying noise to the single-step output of the student model, the multi-step output of the teacher model, and GT (HR image). The first row on the right side of the figure illustrates the difference between the predicted values obtained by inputting GT with added noise into the pre-trained model and the true values. The second and third rows show the score differences predicted by the pre-trained diffusion model after adding noise to the outputs of student model, the outputs of teacher model and GT. Here, we use the symbol ˆ to represent the prediction results of the pre-trained diffuison model after re-adding noise to the model's output.

- We highlight the importance of time in distinguishing between the diffused distributions of real and synthetic data and design a time-aware discriminator to provide efficient and effective supervision for the student model.
- Extensive experiments on real-world SR and blind face restoration (BFR) tasks have demonstrated that our method, using only single-step sampling, achieves performance that is comparable to or surpasses state-of-the-art methods.

## 2 PRELIMINARY

**Diffusion model** is a type of probabilistic generative model, which utilizes a Markov chain to transform complex data distribution $z_0 \sim p_{data}$ into noise distribution $z_T \sim \mathcal{N}(0, I)$ and recover the data by gradually removing the noise. In image super-resolution tasks, Resshift (Yue et al., 2024) changes the initial state of the diffusion model and constructs a new Markov chain to generate high-resolution images. The forward process can be mathematically expressed as follows:

$$q(z_t|z_0, y) = \mathcal{N}\left(z_t|z_0 + \eta_t(z_y - z_0), \kappa^2 \eta_t I\right), \quad (1)$$

where $z_0$ and $z_y$ represent the latent codes obtained by encoding the HR images $x$ and LR images $y$, respectively. $\eta_t$ is a serial of hyper-parameters that monotonically increases with timestep $t$ and satisfies $\eta_0 \to 0$ and $\eta_T \to 1$. $\kappa$ is a hyper-parameter controlling the noise variance. Based on this forward process, the reverse process will commence from the initial state with rich information in low-resolution images to perform denoising. The formula is as follows:

$$q(z_{t-1}|z_t, z_0, y) = \mathcal{N}\left(z_{t-1}|\frac{\eta_{t-1}}{\eta_t}z_t + \frac{\alpha_t}{\eta_t}z_0, \kappa^2 \frac{\eta_{t-1}}{\eta_t}\alpha_t I\right), \quad (2)$$

where $\alpha_t = \eta_t - \eta_{t-1}$. To mitigate the influence of randomness on distillation (Wang et al., 2023c), we reformulate Eq. 2 to employ deterministic sampling as follows:

$$q(z_{t-1}|z_t, z_0, y) = \delta(k_t z_0 + m_t z_t + j_t z_y), \quad (3)$$

where $\delta$ is the unit impulse, $m_t = \sqrt{\frac{\eta_{t-1}}{\eta_t}}$, $j_t = \eta_{t-1} - \sqrt{\eta_{t-1}\eta_t}$ and $k_t = 1 - j_t - m_t$. The details of the derivation can be found in SinSR (Wang et al., 2023c). In the backward process, $z_0$ is usually predicted by a trainable neural network $f_\theta$. The training objective function of $f_\theta$ is as follows:

$$\min_{\boldsymbol{\theta}} \sum_t w_t \|f_{\boldsymbol{\theta}}(\boldsymbol{z}_t, \boldsymbol{y}, t) - \boldsymbol{z}_0\|_2^2, \quad (4)$$

(a) LR    (b) SDS    (c) SDS with HR    (d) SDS with Outputs    (e) HR

Figure 3: **Visualization results with different score distillation techniques.** In the figure, (a) and (e) represent the LR image and its corresponding HR image, respectively. (b) shows the result obtained by employing SDS technique, while (c) and (d) depict the results obtained by leveraging HR images and the output of the teacher model to eliminate bias terms in SDS.

Table 1: **Diffusion-based SR with different score distillation technologies and discriminators on RealSR dataset.** We compare SDS with two score distillation designs that address the inherent biases in SDS. Additionally, based on our proposed score distillation method, we evaluate the performance of a vanilla discriminator, multiple discriminators, and our time-aware discriminator in super-resolution tasks.

| Settings | Score distillation | | | Discriminators | | |
|---|---|---|---|---|---|---|
| | SDS | SDS with HR | SDS with Outputs | Vanilla | Multiple | Time-aware |
| CLIPIQA↑ | 0.450 | 0.556 | **0.671** | 0.711 | 0.724 | **0.741** |
| MUSIQ↑ | 54.069 | 60.079 | **61.506** | 63.550 | 64.223 | **65.701** |

where $w_t = \frac{\alpha_t}{2\kappa^2 \eta_t \eta_{t-1}}$. In practice, omitting this weight often leads to performance improvement.

**Score Distillation Sampling (SDS)** is a distillation technique based on pre-trained diffusion models. It leverages the rich generative prior of pre-trained diffusion models to optimize the generated images or the generator. Specifically, it adds noise to the clean samples generated by the student model and feeds them into a pre-trained diffusion model for prediction. The student model is optimized by calculating the discrepancy between the predicted distribution and the clean sample distribution produced by the student model, which can be expressed as follows:

$$\nabla_\theta \mathcal{L}_{SDS}(z, y, \epsilon, t) = (\epsilon_\phi(z_t, y, t) - \epsilon)\frac{\partial z_t}{\partial \theta}, \tag{5}$$

where $z_t$ refers to the noised version of the clean samples generated by the student model. According to (Poole et al., 2022), the U-Net jacobian term $\frac{\partial \epsilon_\phi(z,y,t)}{\partial z_t}$ is omitted to lead an effective gradient.

## 3 METHODOLOGY

### 3.1 MOTIVATION

Building on prior knowledge of Score Distillation Sampling (SDS), we know that SDS can optimize student models by leveraging the latent knowledge of pre-trained diffusion models, ensuring that the output image distribution aligns as closely as possible with that of the pre-trained diffusion models. However, due to the inherent error in pre-trained diffusion models, we observed that even when GT (HR images) are noised and fed into the pre-trained diffusion model, a deviation still exists between the predicted distribution and the actual data distribution (as illustrated in the first row on the right side of Fig. 2). This indicates that even in ideal situations, SDS itself has biases, consistent with the conclusions of previous related work (Hertz et al., 2023; Wang et al., 2024). Thus, we decompose the gradient calculated by SDS into two components: $\nabla_\theta \mathcal{L}_{SDS} = \epsilon_\phi(z_t^{stu}, y, t) - \epsilon = D_{ir} + \Delta_{bias}$. The first is the expected direction, which guides the student model to generate high-resolution images aligned with the distribution of the teacher model. The second component is the deviation between the predicted and true values of high-quality images that align with the diffusion model's prior distribution. This deviation disrupts the optimization of the student model, producing non-detailed and blurry outputs (as shown in Fig. 3). Our goal is to identify this deviation and eliminate it during the optimization process.

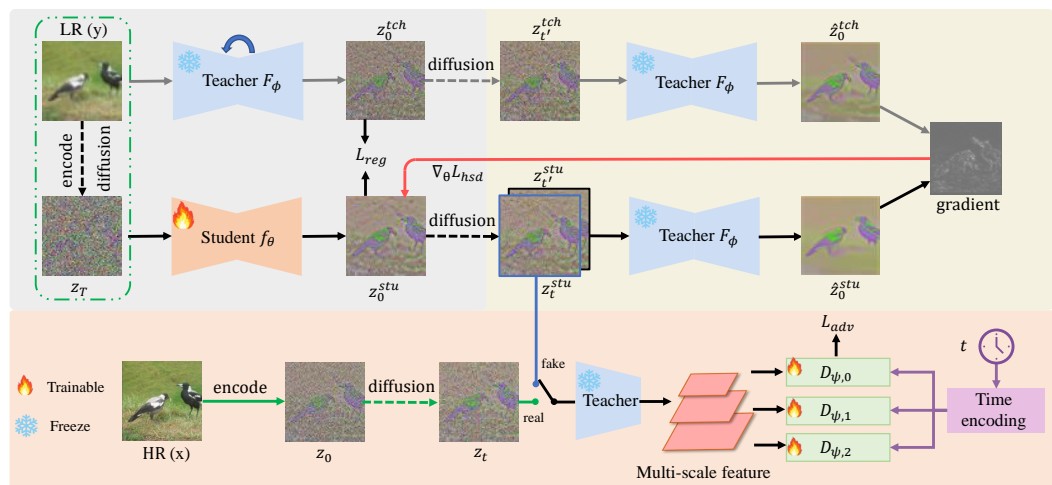

Figure 4: **Method overview.** We train student model to map noisy latent to clean latent through one step sampling. To match the student model's output $z_0^{stu}$ with the multi-step sampling outputs of the teacher model $z_0^{tch}$, we optimize the student model using both regression loss and our proposed HSD. Additionally, to further improve the performance of the student model, we propose a time-aware discriminator that provides effective supervision through adversarial training.

To achieve this, we attempt to re-noise HR image and the output of the teacher model, then input them into the pre-trained diffusion model to calculate the bias $\Delta_{bias} = \epsilon_\phi(\boldsymbol{z}_t, y, t) - \epsilon$ or $\epsilon_\phi(\boldsymbol{z}_t^{tch}, y, t) - \epsilon$. This bias is subsequently subtracted from SDS to guide the model's optimization. The example results are shown in Fig. 3. From the figure, it is evident that the outputs obtained by subtracting the bias using these two methods outperform the results of SDS. Additionally, using the teacher model's output to calculate the score difference and guide the optimization of the student model produces clearer images. This improvement is likely due to the significant difference between HR images and the student model's output, making it challenging to optimize the student model by calculating the score difference on a point-by-point basis. Furthermore, Fig. 2 clearly demonstrates a significant difference in the denoising scores of images generated by the teacher model and the student model under slight noise disturbances (small time steps). Due to the diffusion model's focus on high-frequency information in images at small time steps, it can be concluded that the student model's output notably lacks high-frequency details compared to the teacher model, which aligns with our expectations. Therefore, calculating the score difference between the outputs of the teacher and the student model under mild noise interference provides an effective gradient direction to guide the optimization of the student model.

To ensure that the student model's performance is not overly restricted by the teacher model, we propose incorporating real images into the distillation framework to offer additional supervision. As previously noted, optimizing the model by directly calculating the pixel-wise distance between the real data and the student model's output is difficult. In contrast, we suggest employing adversarial learning to align the output distribution of the student model with that of real data. The successful deployment of pre-trained diffusion models in downstream tasks has revealed that denoising networks can extract multi-level semantic information from images. Consequently, we can utilize the teacher model to extract features and offer supervisory signals to student models via adversarial learning. However, as illustrated in the third row of Fig.2, the distribution difference between the student model's output and the real data varies over time, making it challenging for the discriminator to accurately fit the diffused distribution of the images at different time steps. A straightforward solution is to employ multiple discriminators, each specializing in the diffused distribution at different time steps. As shown in Table 1, this approach significantly enhances the quality of the generated images. However, managing multiple discriminators and their respective time periods introduces complexity and incurs substantial training costs. Given that variations in diffused distribution are primarily related to time steps, we propose that a unique set of parameters can be adaptively learned from each time step and integrated into the discriminator's features. From Table 1, it can be seen that our design effectively improves the quality of generated images.

## 3.2 TAD-SR

The overview framework of our proposed TAD-SR is illustrated in Fig. 4, consisting of a teacher model $F_\phi$ parameterized by $\phi$, a student network $f_\theta$ initialized from the teacher model with weights $\theta$, and a trainable time-aware discriminator $D_\psi$ parameterized by $\psi$. During training, the student model generates samples from noisy data and computes the regression loss against the samples generated iteratively by the teacher model. Subsequently, we introduce slight noise to the samples produced by both the student and teacher models, predict the score function via the teacher model, and refine the student network by leveraging the discrepancy between the two score functions. Furthermore, to mitigate the performance constraints of the teacher model on the student model, we design a time-aware discriminator built upon the encoder network of the pre-trained teacher model, enhancing the perceptual quality of the generated samples through adversarial training processes.

**Regression loss.** We utilize the multi-step output results $z_0^{tch}$ of the teacher model as the learning objective for the student model. It guides the student model to establish a mapping between low-resolution and high-resolution images through single-step inference. The loss is formulated as the following formula:

$$\mathcal{L}_{reg} = \|z_0^{tch} - z_0^{stu}\|_2^2, \quad z_0^{stu} = f_\theta\left(z_T, T, y\right), \tag{6}$$

where $z_T$ is obtained through the forward process Eq. (1). Specifically, Note that our student model samples only the time step $T$ to obtain the noise latent code $z_T \sim \mathcal{N}\left(x_t; y, \kappa^2 \eta_t \boldsymbol{I}\right)$.

**High-frequency enhanced score distillation.** As analyzed in Section 3.1, employing SDS (Poole et al., 2022) to accelerate diffusion-based SR models is not an optimal solution. Its inherent bias may introduce meaningless gradient directions to the student model, leading to a blurring and smoothing output (Wang et al., 2024; Hertz et al., 2023). To eliminate this bias, DMD (Yin et al., 2023) trains a new diffusion model to learn the score function of samples generated by the student model and updates the generator based on the difference between the score functions predicted by the new model and the teacher model. However, this approach involves a complex training process that requires alternating training between the student model and the new diffusion model.

By contrast, based on the observations presented in Fig. 2, we develop an effective and efficient score distillation method. Specifically, we calculate the difference between the predicted score function of the teacher model's output and the true score function to obtain the bias term in score distillation sampling. By subtracting this bias term, we obtain a meaningful gradient direction. According to Eq. 5, the following formula is derived:

$$\mathcal{L}_{hsd} = \mathbb{E}_{z_{t'}^{tch}, z_{t'}^{stu}, y}\left[\omega\left(\left(\epsilon_\phi\left(z_{t'}^{stu}, t, y\right) - \epsilon\right) - \left(\epsilon_\phi\left(z_{t'}^{tch}, t, y\right) - \epsilon\right)\right)\right], \tag{7}$$

where $\omega = 1/CS$ is a weighting function, $C$ is the number of channels and $S$ is the number of spatial pixels. $z_{t'}^{tch}$ and $z_{t'}^{stu}$ are the noise data obtained by adding noise to the outputs of the teacher model $z_0^{tch}$ and the output of student model $z_0^{stu}$, respectively, through Eq. 1. By simplifying this formula, our high-frequency enhanced score distillation (HSD) technique essentially calculates the score difference $\epsilon_\phi\left(z_{t'}^{stu}, t, y\right) - \epsilon_\phi\left(z_{t'}^{tch}, t, y\right)$ between the teacher model and the student model's outputs under different degrees of noise interference. As can be seen from the second row of Fig. 2, these differences are primarily significant under mild noise disturbances (*i.e.,* small time steps). Given that diffusion models typically predict high-frequency information in images at small time steps, this suggests that images generated by student models are predominantly deficient in high-frequency details compared to those produced by teacher models. Consequently, we mainly constrain the score difference between the student model and the teacher model output under slight noise disturbance, specifically when $t' \sim U\left(1, T/5\right)$. According to Eq. 1, we can simplify Eq. 7 as follows:

$$\mathcal{L}_{hsd} = \mathbb{E}_{z_{t'}^{tch}, z_{t'}^{stu}, y}\left[\omega_2\left(z_0^{stu} - z_0^{tch} + F_\phi\left(z_{t'}^{tch}, t, y\right) - F_\phi\left(z_{t'}^{stu}, t, y\right)\right)\right], \tag{8}$$

where $\omega_2 = \frac{\omega\left(1 - \eta_{t'}\right)}{\sqrt{\eta_{t'}}\kappa}$. The details of the derivation can be found in the appendix. Note that during the loss backpropagation in Eq. 7, similar to SDS, we omit the U-Net Jacobian matrix term.

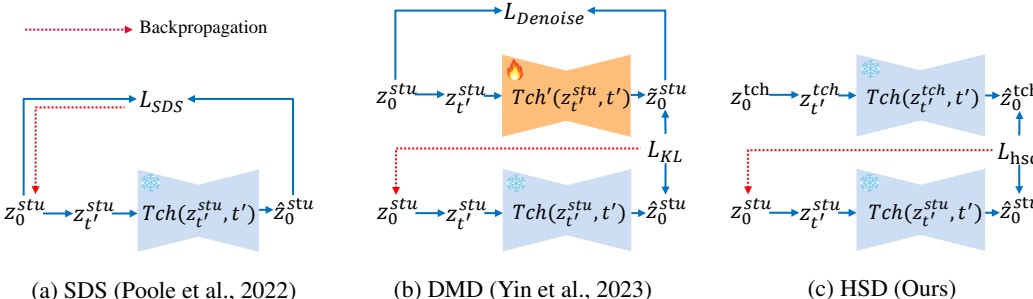

Figure 5: **Comparison of various score distillation techniques.** Compared to SDS (Poole et al., 2022; Sauer et al., 2023) and DMD (Yin et al., 2023), our high-frequency enhanced score distillation fully utilizes the potential of teacher model, providing meaningful gradient guidance to student models without training an extra diffusion model.

From the above equation, it can be seen that when the output of the student model is the same as that of the teacher model, the loss is zero, and there is no additional bias. Compared to SDS, our proposed HSD provides more meaningful gradient guidance for student models.

**Time-aware discriminator.** To prevent the student model's performance from being entirely constrained by the teacher model, we propose incorporating real images (HR images) into the distillation framework. However, directly calculating the regression loss between the real image and the student model's output can result in optimization challenges. Recent studies (Sauer et al., 2023) (Sauer et al., 2024) have shown that adversarial loss can be integrated into diffusion models to enhance the quality of generated images. However, ADD (Sauer et al., 2023) relies on pre-training the DINOv2 discriminator in pixel space, which is both costly and complex. To reduce training costs and enhance model performance, LADD (Sauer et al., 2024) employed a pre-trained diffusion model for adversarial training in latent space. Despite its contribution, LADD overlooks the critical correlation between the features extracted by the diffusion model and their corresponding time steps. It relies on a single discriminator to differentiate between the distribution differences of real and synthetic data under various noise disturbances, which poses significant challenges for optimizing the discriminator. To address this issue, we propose a time-aware discriminator, which is capable of distinguishing between the distributions of real and generated images that have undergone various perturbations in latent space. Specifically, we first utilize the encoder part of the teacher model to extract multi-scale features $F_k$ from both the student model's output images and real images.

$$F_k = Enc_\phi(z_t, t, y),$$ (9)

where $Enc_\phi$ denotes the encoder part of the teacher model's denoising network, $k$ denotes the scale of the extracted features. $z_t$ represents the noisy latent code after adding noise to the real latent code. We use $F_k^{stu}$ to denote the multi-scale features extracted from the output of the student model. We then encode the time step $t$ as of sinusoidal timestep embeddings, which are sent to different discriminator heads $D_{\psi,k}$ to learn a set of parameters $\gamma_k$ and $\beta_k$ through several linear layers. These parameters are used to modulate multi-scale features: $Norm(F_k) * (1 + \gamma_k) + \beta_k$.

After modulation, the features at each scale are evaluated through their corresponding discriminator heads. The final output is obtained by averaging the results from each discriminator head. For simplicity, we denote the process of modulating and discriminating features in the discriminator head as $D_{\psi,k}(F_k, t)$. Consequently, the corresponding adversarial loss can be formulated as follows:

$$\mathcal{L}_{adv}^{f_\theta} = -\mathbb{E}_{z_0^{stu}}\left[\sum_k D_{\psi,k}\left(F_k^{stu}, t\right)\right],$$ (10)

$$\mathcal{L}_{adv}^{D_\psi} = \mathbb{E}_{z_0^{stu}}\left[\sum_k \max\left(0, 1 + D_{\psi,k}\left(F_k^{stu}, t\right)\right)\right] + \mathbb{E}_{z_0}\left[\sum_k \max\left(0, 1 - D_{\psi,k}\left(F_k, t\right)\right)\right].$$ (11)

Table 2: Quantitative results of different methods on the dataset of *ImageNet-Test*. The best and second best results are highlighted in **bold** and underline. ∗ indicates that the result was obtained by replicating the method in the paper.

| Methods | Metrics | | | | |
|---|---|---|---|---|---|
| | PSNR↑ | SSIM↑ | LPIPS↓ | CLIPIQA↑ | MUSIQ↑ |
| ESRGAN | 20.67 | 0.448 | 0.485 | 0.451 | 43.615 |
| RealSR-JPEG | 23.11 | 0.591 | 0.326 | 0.537 | 46.981 |
| BSRGAN | 24.42 | 0.659 | 0.259 | 0.581 | 54.697 |
| SwinIR | 23.99 | 0.667 | 0.238 | 0.564 | 53.790 |
| RealESRGAN | 24.04 | 0.665 | 0.254 | 0.523 | 52.538 |
| DASR | 24.75 | 0.675 | 0.250 | 0.536 | 48.337 |
| LDM-15 | 24.89 | 0.670 | 0.269 | 0.512 | 46.419 |
| ResShift-15 | **25.01** | **0.677** | 0.231 | 0.592 | 53.660 |
| SinSR-1 | 24.56 | 0.657 | **0.221** | 0.611 | 53.357 |
| SinSR*-1 | 24.59 | 0.659 | 0.231 | 0.599 | 52.462 |
| DMD*-1 | 24.05 | 0.629 | 0.246 | 0.612 | 54.124 |
| *TAD-SR-1* | 23.91 | 0.641 | 0.227 | **0.652** | **57.533** |

Table 3: Quantitative results of different methods on two real-world datasets.

| Methods | Datasets | | | |
|---|---|---|---|---|
| | *RealSR* | | *RealSet65* | |
| | CLIPIQA↑ | MUSIQ↑ | CLIPIQA↑ | MUSIQ↑ |
| ESRGAN | 0.236 | 29.048 | 0.374 | 42.369 |
| RealSR-JPEG | 0.362 | 36.076 | 0.528 | 50.539 |
| BSRGAN | 0.543 | 63.586 | 0.616 | 65.582 |
| SwinIR | 0.465 | 59.636 | 0.578 | 63.822 |
| RealESRGAN | 0.490 | 59.678 | 0.600 | 63.220 |
| DASR | 0.363 | 45.825 | 0.497 | 55.708 |
| LDM-15 | 0.384 | 49.317 | 0.427 | 47.488 |
| ResShift-15 | 0.596 | 59.873 | 0.654 | 61.330 |
| SinSR-1 | 0.689 | 61.582 | 0.715 | 62.169 |
| SinSR*-1 | 0.691 | 60.865 | 0.712 | 62.575 |
| DMD*-1 | 0.709 | 63.610 | 0.723 | 66.177 |
| *TAD-SR-1* | **0.741** | **65.701** | **0.734** | **67.500** |

**The total objective.** The student network is trained with the above three losses as follows:

$$\mathcal{L}_{f_\theta} = \mathcal{L}_{reg} + \lambda_1 \mathcal{L}_{hsd} + \lambda_2 \mathcal{L}_{adv}^{f_\theta}, \tag{12}$$

where $\lambda_1$ and $\lambda_2$ are the hyperparameters to control the relative importance of these objectives.

# 4 EXPERIMENTS

## 4.1 EXPERIMENTAL SETUP

**Training Details.** For a fair comparison, we follow the same experimental setup and backbone design as that in (Yue et al., 2024; Wang et al., 2023c). Specifically, we use the weights of the teacher model (ResShift) to initialize the student model, and then train the model for 30K iterations based on our proposed loss functions. For real-world SR task, we set the weighting factor $\lambda_1 = 1$ and $\lambda_2 = 0.02$. For blind face restoration (BFR) task, we set $\lambda_1 = 0.1$ and $\lambda = 0.2$.

**Compared methods.** For real-world SR task, we evaluate the effectiveness and efficiency of TAD-SR in comparison to representative SR models, including BSRGAN (Zhang et al., 2021), SwinIR (Liang et al., 2021), RealESRGAN (Wang et al., 2021b), DASR (Liang et al., 2022b), RealSR-JPEG (Ji et al., 2020) LDM (Rombach et al., 2022), ResShift (Yue et al., 2024) and SinSR (Wang et al., 2023c). Additionally, we also apply DMD (Yin et al., 2023) to super-resolution tasks as a baseline. For BFR task, we compare TAD-SR with recent BFR methods, including DFDNet (Li et al., 2020), PSFRGAN (Chen et al., 2021), GFPGAN (Wang et al., 2021a), RestoreFormer (Wang et al., 2022), VQFR (Gu et al., 2022), CodeFormer (Zhou et al., 2022), and DifFace (Yue & Loy, 2022).

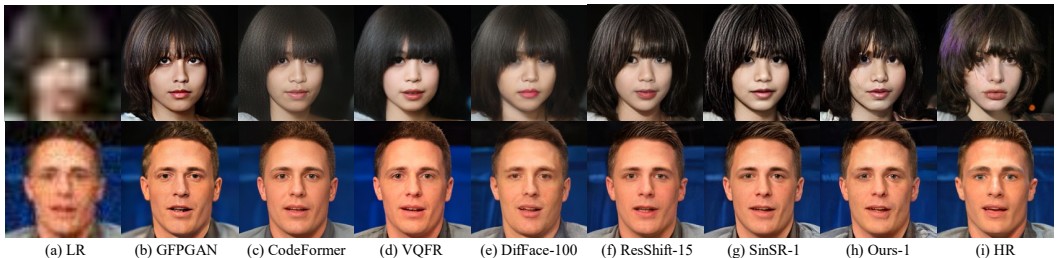

(a) LR    (b) SwinIR    (c) BSRGAN    (d) LDM-15    (e) ResShift-15    (f) DMD-1    (g) SinSR-1    (h) Ours-1    (i) HR

Figure 6: Qualitative comparisons of different methods on two synthetic examples of the *ImageNet-Test* dataset. Please zoom in for a better view.

(a) LR    (b) GFPGAN    (c) CodeFormer    (d) VQFR    (e) DifFace-100    (f) ResShift-15    (g) SinSR-1    (h) Ours-1    (i) HR

Figure 7: Qualitative comparisons of different methods on two synthetic examples of the *CelebA-Test* dataset. Please zoom in for a better view.

**Metrics.** For real-world SR tasks, we utilize LPIPS (Zhang et al., 2018b), CLIPIQA (Wang et al., 2023a) and MUSIQ (Ke et al., 2021) as evaluation metrics. PSNR and SSIM (Wang et al., 2004) are also reported for reference. For BFR task, we also evaluate methods with identity score (IDS), landmark distance (LMD) and FID (Heusel et al., 2017). Note that we take non-reference metrics as the primary metrics since they are closer to human perception (Wang et al., 2023b; Xie et al., 2024).

**Datasets.** For the real-world image super-resolution task, we train the models on the training set of ImageNet (Deng et al., 2009) following the same pipeline with ResShift (Yue et al., 2024) where the degradation model is adopted from RealESRGAN (Wang et al., 2021b). Then, we evaluate our model on one synthetic dataset ImageNet-Test (Deng et al., 2009; Yue et al., 2024) and two real-word datasets RealSR (Cai et al., 2019) and RealSet65 (Yue et al., 2024). For the BFR task, We train the models on FFHQ dataset (Karras et al., 2019), and the LQ images are synthesized following a typical degradation model used in (Wang et al., 2021a). One synthetic dataset CelebA-Test (Karras et al., 2018; Yue et al., 2024) and three real-world datasets LFW (Huang et al., 2008), WebPhoto and WIDER (Yang et al., 2016) are adopted to evaluate the performance of face restoration model.

### 4.2 EXPERIMENTAL RESULTS

**Evaluation on synthetic datasets.** For the real-world SR task, we conduct a comprehensive comparison between TAD-SR and other SR methods on the ImageNet-Test dataset, as summarized in Table 2 and Fig. 6. The following conclusions can be drawn: i) TAD-SR significantly outperforms other methods in terms of non-reference metrics, and achieves second-best results in the full-reference metric LPIPS. It demonstrates that TAD-SR has the ability to generate images with high perceptual quality and realism. ii) Visual results show that TAD-SR produces images with higher clarity and better visual perception. Additionally, the complexity comparison of different SR methods is presented in Table 6. The table shows that our method improves the inference speed of the teacher model by approximately tenfold. For BFR task, We used CelebA-Test as the testing dataset, and the results are summarized in Table 4 and Fig. 7. From the perspective of evaluation metrics, the proposed method achieves SOTA results in terms of FID and comparable results in terms of IDS, LMD, CLIPIQA, and MUSIQ, which demonstrates the effectiveness of TAD-SR on BFR task. As shown in Fig. 7, the generated faces by TAD-SR appear more natural and exhibit richer details. Furthermore, we visualize the spectrograms obtained from the Fourier transform of images generated by TAD-SR and other methods. As shown in Fig. 10, the spectrograms indicate that TAD-SR retains more high-frequency information compared to other methods.

Table 4: Quantitative results of different methods on the dataset of *CelebA-Test*. The best and second best results are highlighted in **bold** and underline.

| Methods | Metrics | | | | | | |
|---|---|---|---|---|---|---|---|
| | LPIPS↓ | IDS↓ | LMD↓ | FID-F↓ | FID-G↓ | CLIPIQA↑ | MUSIQ↑ |
| DFDNet | 0.739 | 86.323 | 20.784 | 93.621 | 76.118 | 0.619 | 51.173 |
| PSFRGAN | 0.475 | 74.025 | 10.168 | 63.676 | 60.748 | 0.630 | 69.910 |
| GFPGAN | 0.416 | 66.820 | 8.886 | 66.308 | 27.698 | 0.671 | 75.388 |
| RestoreFormer | 0.488 | 70.518 | 11.137 | 50.165 | 51.997 | **0.736** | 71.039 |
| VQFR | 0.411 | 65.538 | 8.910 | 58.423 | 25.234 | 0.685 | 73.155 |
| CodeFormer | 0.324 | **59.136** | 5.035 | 62.794 | 26.160 | 0.698 | **75.900** |
| DifFace-100 | 0.338 | 63.033 | 5.301 | 52.531 | 23.212 | 0.527 | 66.042 |
| ResShift-4 | **0.309** | 59.623 | 5.056 | 50.164 | 17.564 | 0.613 | 73.214 |
| SinSR*-1 | 0.319 | 60.305 | **4.935** | 55.292 | 21.681 | 0.634 | 74.140 |
| *TAD-SR-1* | 0.341 | 59.897 | 5.050 | **41.968** | **16.779** | 0.735 | 75.027 |

Table 5: Quantitative results of different methods on three real-world human face datasets.

| Methods | Datasets | | | | | |
|---|---|---|---|---|---|---|
| | *LFW* | | *WebPhoto* | | *WIDER* | |
| | CLIPIQA↑ | MUSIQ↑ | CLIPIQA↑ | MUSIQ↑ | CLIPIQA↑ | MUSIQ↑ |
| DFDNet | 0.716 | 73.109 | 0.654 | 59.024 | 0.625 | 63.210 |
| PSFRGAN | 0.647 | 73.602 | 0.637 | 71.674 | 0.648 | 71.507 |
| GFPGAN | 0.687 | 74.836 | 0.651 | 73.369 | 0.663 | **74.694** |
| RestoreFormer | 0.741 | 73.704 | 0.709 | 69.837 | 0.730 | 67.840 |
| VQFR | 0.710 | 74.386 | 0.677 | 70.904 | 0.707 | 71.411 |
| CodeFormer | 0.689 | **75.480** | 0.692 | **74.004** | 0.699 | 73.404 |
| DifFace-100 | 0.593 | 70.362 | 0.555 | 65.379 | 0.561 | 64.970 |
| ResShift-4 | 0.626 | 70.643 | 0.621 | 71.007 | 0.629 | 71.084 |
| SinSR*-1 | 0.640 | 72.457 | 0.641 | 73.357 | 0.654 | 73.556 |
| *TAD-SR-1* | **0.768** | 74.085 | **0.718** | 71.952 | **0.770** | 73.739 |

**Evaluation on real-world datasets.** In addition to evaluating our method on synthetic datasets, we also assess the method in real-world datasets. As shown in Table 3, in terms of non-reference metrics, the proposed method significantly outperforms other methods with just a single-step sampling. Specifically, when compared to ResShift, which serves as our teacher model, the non-reference metrics show substantial improvement after applying TAD-SR. Additionally, visual comparisons are displayed in Fig 1 and Fig. 11. To ensure a comprehensive evaluation, we include diverse scenarios, such as buildings, animals, and landscapes. It can be observed that the images generated by TAD-SR appear more naturalistic, as evidenced by the distinct brick textures, as well as the fine and natural-looking polar bear fur. For BFR task, we evaluate recent methods on LFW, Webphoto, and WIDER datasets. The results are presented in Table 5, leading to several significant conclusions. Across all three datasets, the proposed method achieves the highest CLIPIQA, outperforming other methods by a substantial margin. On the WIDER dataset, the proposed method also achieves the second-best MUSIQ. All these results inform that in terms of BFR task, TAD-SR can generate images with really high perceptual quality. Visual comparisons are provided in Fig. 14, where it is evident that TAD-SR produces more realistic hair details, sharper facial contours, and improved skin textures.

## 5 CONCLUSION

In this paper, we propose a time-aware distillation method that accelerates diffusion-based super-resolution models to a single inference step. We introduce a high-frequency enhanced score distillation technique that optimizes the generator by calculating the score difference between the outputs of the teacher and student models following slight noise perturbation, thereby enhancing the high-frequency details in the student model's output. To elevate the student model's performance ceiling, we incorporate generative adversarial learning into the diffusion model framework. Specifically, we design a time-aware discriminator that distinguishes between generated and real data in latent space, providing more efficient and effective supervision for the student model. Extensive experiments demonstrate that our method can achieve performance on par with or surpassing that of the SOTA methods in a single inference step.

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

# A APPENDIX

## A.1 RELATED WORK

### A.1.1 IMAGE SUPER-RESOLUTION.

Traditional methods (Dong et al., 2012; Gu et al., 2017; 2015) for image super-resolution rely on manual design of image priors based on subjective knowledge to restore image details. With the advancement of deep learning (DL), DL-based image super-resolution has become predominant, which mainly focuses on network architecture (Lai et al., 2017; Menick & Kalchbrenner, 2018; Lugmayr et al., 2020; Sajjadi et al., 2017), image priors (Pan et al., 2021; Chan et al., 2021), loss functions (Zhou et al., 2020; Fuoli et al., 2021), and other aspects (Zhang et al., 2018a; Wang et al., 2021b). Recently, diffusion-based methods for image super-resolution have garnered widespread attention. SR3 (Saharia et al., 2022) incorporated low-resolution images as conditions into the denoising model to guide the sampling process. Subsequently, CDPMSR (Niu et al., 2023) and IDM (Gao et al., 2023) respectively utilized preprocessed images and features as conditions to enhance the perceptual quality. Inspired by the powerful generation priors of stable diffusion (SD) (Rombach et al., 2022), recent studies (Wang et al., 2023b; Yang et al., 2023; Wu et al., 2024b) have achieved image super-resolution by fine-tuning pre-trained SD models (Rombach et al., 2022). However, these methods typically require dozens or even hundreds of iterations to generate high-resolution images. To enhance the inference efficiency, ResShift (Yue et al., 2024) redesigned the diffusion process by shifting the residuals between high-resolution and low-resolution images to construct a Markov chain, achieving performance comparable to previous state-of-the-art methods with just 15 sampling steps. During the same period as our method, OSEDiff (Wu et al., 2024a) directly utilized LR images as the starting point for diffusion and optimized the student model through variational score distillation, generating HR images through a single sampling step. However, it relies on a specific model architecture, while our approach offers a more generalized method for accelerating diffusion models, enabling the distillation of various super-resolution models into single-step sampling based on specific requirements. Furthermore, our method can theoretically be extended to other tasks, such as unconditional generation.

### A.1.2 ACCELERATING DIFFUSION MODELS.

Although diffusion model (Ho et al., 2020; Rombach et al., 2022) has formidable generation capabilities, the substantial number of inference steps poses a significant obstacle to its practical implementation. Recent studies focusing on enhancing the inference speed of diffusion models have garnered considerable interest within the research community. Mainstream approaches include the development of high-order samplers (Song et al., 2020; Lu et al., 2022; Zheng et al., 2024) and the application of knowledge distillation techniques (Salimans & Ho, 2022; Sauer et al., 2023; 2024; Song et al., 2023; Luo et al., 2023). Denoising diffusion implicit models (DDIM) (Song et al., 2020), an early contribution, introduced a deterministic sampling method that notably decreased the number of diffusion sampling steps. DPMSolver (Lu et al., 2022) proposed a fast dedicated high-order ODE solver, further reducing the diffusion sampling steps to 20. However, trajectory compression through numerical solvers often results in performance degradation, necessitating over ten inference steps to generate samples. In contrast, progressive distillation (Salimans & Ho, 2022) gradually reduces the inference steps of student models through multi-stage distillation, but the accumulation of errors in each distillation stage may affect the performance of the student model. Consistency model (Song et al., 2023) eliminates the need for computation-intensive iterations by

Table 6: Complexity comparison among different SR methods. All methods are tested on the $\times 4$ (64$\rightarrow$256) SR tasks, and the inference time is measured on an A100 GPU.

| Method | LDM | ResShift | SinSR | DMD* | TAD-SR |
|---|---|---|---|---|---|
| NFE | 15 | 15 | 1 | 1 | 1 |
| Inference time (s) | 0.408 | 0.682 | 0.058 | 0.058 | 0.058 |
| #Params (M) | 168.92 | 173.91 | 173.91 | 173.91 | 173.91 |

applying consistency regularization to ODE trajectories. Additionally, Adversarial diffusion distillation (ADD) (Sauer et al., 2023) integrates generative adversarial networks with score distillation to enhance the perceptual quality of student network-generated images. For image super-resolution tasks, AddSR (Xie et al., 2024) introduces two key advancements based on adversarial distillation technology, effectively fulfilling image super-resolution objectives. Inspired by cycle consistency loss, SinSR (Wang et al., 2023c) proposes a single-step image super-resolution method. However, AddSR overlooks the influence of time steps on the discriminator, while SinSR primarily focuses on constraining latent codes through pixel-level loss, neglecting perceptual distribution alignment. To achieve image super-resolution more efficiently and effectively, this work proposes a time-aware diffusion distillation method.

## A.2 IMPLEMENTATION DETAILS

### A.2.1 MATHEMATICAL DETAILS

- **Derivation of Eq. equation 8**: According to the transition distribution of Eq. equation 1 of our manuscript, the predicted noise $\epsilon_\phi$ can be expressed via the following reparameterization trick:

$$\epsilon_\phi = \frac{z_t - (\hat{z}_0 + \eta_t (z_y - \hat{z}_0))}{\sqrt{\eta_t \kappa}},\tag{13}$$

where $\hat{z}_0 = F_\phi(z_t, t, y)$. According to the Eq. equation 13, we can rewrite Eq. equation 7 as follows:

$$\mathcal{L}_{hsd} = \mathbb{E}_{z_{t'}^{tch}, z_{t'}^{stu}, y} \left[ \frac{\omega \left( \left( z_{t'}^{stu} - z_{t'}^{tch} \right) + (1 - \eta_{t'}) \left( F_\phi \left( z_{t'}^{tch}, t', y \right) - F_\phi \left( z_{t'}^{stu}, t', y \right) \right) \right)}{\sqrt{\eta_{t'} \kappa}} \right].\tag{14}$$

Since the noise injected into the output image of the student model and the output image of the teacher model is the same, we have: $z_{t'}^{stu} - z_{t'}^{tch} = (1 - \eta_{t'}) \left( z_0^{stu} - z_0^{tch} \right)$. Then Eq. equation 14 can be written as:

$$\mathcal{L}_{hsd} = \mathbb{E}_{z_{t'}^{tch}, z_{t'}^{stu}, y} \left[ \frac{\omega (1 - \eta_{t'}) \left( z_0^{stu} - z_0^{tch} + F_\phi \left( z_{t'}^{tch}, t', y \right) - F_\phi \left( z_{t'}^{stu}, t', y \right) \right)}{\sqrt{\eta_{t'} \kappa}} \right]$$
$$= \mathbb{E}_{z_{t'}^{tch}, z_{t'}^{stu}, y} \left[ \omega_2 \left( z_0^{stu} - z_0^{tch} + F_\phi \left( z_{t'}^{tch}, t', y \right) - F_\phi \left( z_{t'}^{stu}, t', y \right) \right) \right],\tag{15}$$

where $\omega_2 = \frac{\omega (1 - \eta_{t'})}{\sqrt{\eta_{t'} \kappa}}$

### A.2.2 TAD-SR TRAINING PROCEDURE

For a comprehensive understanding, we provide a detailed description of our TAD-SR training procedure in Algorithm 1.

---

**Algorithm 1:** TAD-SR Training Procedure

---

**Input:** Pretrained diffusion model $F_\phi$, paired dataset $\mathcal{D} = \{x, y\}$, Time steps $T$
**Output:** Trained generator $f_\theta$ and disriminator $D_\psi$.

1  // Initialize generator from pretrained model
2  $f_\theta \leftarrow \text{copyWeights}(F_\Phi)$,
3  **while** *train* **do**
4      // Generated images
5      Sample $\epsilon \sim \mathcal{N}(0, \mathbf{I})$, $(x, y) \sim \mathcal{D}$
6      $z_T \leftarrow \text{Forward process}(T, y, x, \epsilon)$ // Eq 1
7      $z_0^{stu} \leftarrow f_\theta(z_T, y, T)$ // One-step
8      $z_0^{tch} \leftarrow F_\phi(z_T, y, T)$ // Multi-step
9
10     // Update discriminator model
11     Sample time step $t \sim \mathcal{U}(0, T)$
12     $z_t^{stu} \leftarrow \text{Forward process}(t, y, z_0^{stu}, \epsilon)$ // Eq 1
13     $z_t \leftarrow \text{Forward process}(t, y, z_0, \epsilon)$ // Eq 1
14     $\mathcal{L}_{\text{adv}}^{D_\psi} \leftarrow \text{Adversarial loss}(z_t^{stu}, z_t, y, t)$ // Eq 9 and Eq 11
15     $D_\psi \leftarrow \text{update}(D_\psi, \mathcal{L}_{\text{adv}}^{D_\psi})$
16
17     // Update generator
18     Sample $\epsilon^{'} \sim \mathcal{N}(0, \mathbf{I}), t^{'} \sim \mathcal{U}(0, T/5)$
19     $z_{t'}^{stu} \leftarrow \text{Forward process}(t^{'}, y, z_0^{stu}, \epsilon^{'})$ // Eq 1
20     $z_{t'}^{tch} \leftarrow \text{Forward process}(t^{'}, y, z_0^{tch}, \epsilon^{'})$ // Eq 1
21     $\mathcal{L}_{\text{reg}} \leftarrow \text{Regression loss}(z_0^{stu}, z_0^{tch})$ // Eq 6
22     $\mathcal{L}_{\text{hsd}} \leftarrow \text{HSD}(z_{t'}^{stu}, z_{t'}^{tch}, y, t^{'})$ // Eq 7
23     $\mathcal{L}_{\text{adv}}^{f_\theta} \leftarrow \text{Adversarial loss}(z_t^{stu}, y, t)$ // Eq 9 and Eq 10
24     $\mathcal{L}_{f_\theta} \leftarrow \mathcal{L}_{\text{reg}} + \lambda_1 \mathcal{L}_{\text{hsd}} + \lambda_2 \mathcal{L}_{\text{adv}}^{f_\theta}$
25     $f_\theta \leftarrow \text{update}(f_\theta, \mathcal{L}_{f_\theta})$
26 **end while**

---

Table 7: Ablation studies of the proposed methods on *ImageNet-Test* benchmark. The best results are highlighted in **bold**.

| Score distillation | Discriminator | PSNR↑ | SSIM↑ | LPIPS↓ | CLIPIQA↑ | MUSIQ↑ |
|---|---|---|---|---|---|---|
| SDS | ✗ | 24.46 | 0.658 | 0.335 | 0.412 | 41.133 |
| SDS | ✓ | **24.76** | 0.670 | 0.300 | 0.469 | 46.024 |
| SDS | time-aware | 24.69 | **0.671** | 0.278 | 0.522 | 49.932 |
| HSD | ✗ | 24.64 | 0.661 | 0.228 | 0.608 | 53.508 |
| HSD | ✓ | 23.89 | 0.640 | **0.227** | 0.649 | 57.370 |
| HSD | time-aware | 23.91 | 0.641 | **0.227** | **0.652** | **57.533** |

Table 8: Ablation studies of the proposed methods on *RealSR* and *RealSet65* benchmarks. The best results are highlighted in **bold**.

| Score distillation | Discriminator | *RealSR/RealSet65* | |
|---|---|---|---|
| | | CLIPIQA↑ | MUSIQ↑ |
| SDS | ✗ | 0.450/0.484 | 54.069/52.923 |
| SDS | ✓ | 0.489/0.528 | 57.290/57.567 |
| SDS | time-aware | 0.538/0.554 | 60.223/59.627 |
| HSD | ✗ | 0.671/0.697 | 61.506/63.609 |
| HSD | ✓ | 0.711/0.729 | 63.550/66.904 |
| HSD | time-aware | **0.741/0.734** | **65.701/67.500** |

## A.3 ADDITIONAL EXPERIMENTS

### A.3.1 ABLATION STUDY

The aforementioned experiments have confirmed the effectiveness of our method in image super-resolution tasks. This section is dedicated to presenting ablation studies that aim to further validate the importance of the crucial modules introduced within our framework.

**High-frequency enhanced score distillation.** We first investigate the importance of high-frequency enhanced score distillation. Recall that in Section 3.2, we analyzed how high-frequency enhanced score distillation can provide meaningful guidance for optimizing student model compared to score distillation sampling (SDS). Here, we further validate its effectiveness through experiments. As shown in Table 8 and Table 7, compared with SDS, our proposed high-frequency enhanced score distillation (HSD) can significantly improve the LPIPS, CILIPIQA and MUSIQ scores on all datasets. Additionally, with the introduction of adversarial learning, HSD also achieves superior metrics compared to SDS, further validating that the proposed method enhances image generation quality and surpasses SDS.

**Time-aware discriminator.** It has been proven that introducing generative adversarial training in latent space is easier to optimize and more cost-effective than pixel space (Sauer et al., 2024). Now, we demonstrate the importance of introducing time injection into the discriminator. Intuitively, when the discriminator does not have time injection, it needs to distinguish the distribution between real data and generated data under different noise disturbances, which is undoubtedly extremely challenging. Adding time injection to the discriminator is equivalent to providing additional information related to the level of noise disturbance, which should improve the performance of the discriminator and provide more effective supervision for the generator. We further validated the above analysis through experiments. As shown in Table 8, performance improves with the replacement of the standard discriminator by our proposed time-aware discriminator, regardless of the score distillation technique used. We also conduct ablation experiments to evaluate the impact of using multi-scale

Table 9: Ablation studies of the proposed discriminator on *RealSR* and *RealSet65* benchmarks. The best results are highlighted in **bold**.

| Discriminator | *RealSR* | | *RealSet65* | |
|---|---|---|---|---|
| | CLIPIQA↑ | MUSIQ↑ | CLIPIQA↑ | MUSIQ↑ |
| Ours | **0.741** | **65.701** | **0.734** | **67.500** |
| w/o time-aware | 0.711 | 63.550 | 0.729 | 66.904 |
| w/o multi-scale | 0.722 | 65.205 | 0.724 | 67.330 |

Table 10: Performance comparison of the proposed high-frequency enhanced score distillation techniques across varying time-period sampling lengths.

| Time-period lengths | Datasets | | | |
|---|---|---|---|---|
| | RealSR | | RealSet65 | |
| | CLIPIQA↑ | MUSIQ↑ | CLIPIQA↑ | MUSIQ↑ |
| T/5 | **0.741** | **65.701** | **0.734** | **67.500** |
| 2T/5 | 0.730 | 65.223 | 0.732 | 67.292 |
| 3T/5 | 0.731 | 65.431 | 0.730 | 67.254 |
| 4T/5 | 0.731 | 65.122 | 0.731 | 67.263 |
| T | 0.733 | 65.321 | 0.731 | 67.303 |

Table 11: Quantitative comparison with state of the arts on RealSR dataset dataset. The best and second best results are highlighted in **bold** and underline.

| Methods | RealSR | | | | | | |
|---|---|---|---|---|---|---|---|
| | PSNR↑ | LPIPS↓ | FID↓ | NIQE↓ | CLIPIQA↑ | MUSIQ↑ | MANIQA↑ |
| BSRGAN | 26.49 | **0.267** | 141.28 | 5.66 | 0.512 | 63.28 | 0.376 |
| RealESRGAN | 25.78 | 0.273 | 135.18 | 5.83 | 0.449 | 60.36 | 0.373 |
| LDL | 25.09 | 0.277 | 142.71 | 6.00 | 0.430 | 58.04 | 0.342 |
| FeMaSR | 25.17 | 0.294 | 141.05 | 5.79 | 0.541 | 59.06 | 0.361 |
| StableSR-200 | 25.63 | 0.302 | 133.40 | 5.76 | 0.528 | 61.11 | 0.366 |
| ResShift-15 | 26.34 | 0.346 | 149.54 | 6.87 | 0.542 | 56.06 | 0.375 |
| PASD-20 | **26.67** | 0.344 | 122.30 | 6.06 | 0.519 | 62.92 | 0.404 |
| SeeSR-50 | 25.24 | 0.301 | 125.42 | 5.39 | 0.670 | **69.82** | **0.540** |
| +UniPC-10 | 25.86 | 0.281 | 122.41 | 5.53 | 0.577 | 67.12 | 0.476 |
| +DPMSolver-10 | 25.90 | 0.281 | 122.46 | 5.54 | 0.581 | 67.12 | 0.478 |
| SinSR-1 | 26.16 | 0.308 | 142.44 | 5.75 | 0.630 | 60.96 | 0.399 |
| AddSR-1 | 23.12 | 0.309 | 132.01 | 5.54 | 0.552 | 67.14 | 0.488 |
| OSEDiff-1 | 25.15 | 0.292 | 123.49 | 5.63 | 0.668 | 68.99 | 0.474 |
| TAD-SR-1 | 24.50 | 0.304 | **118.38** | **5.13** | **0.676** | 69.02 | 0.526 |

features in the discriminator. We designed an experiment using only the features of the last layer of the diffusion model for discrimination, denoted as "w/o multi-scale". From Table 9, it can be seen that the discriminator utilizing multi-scale features and incorporating temporal information achieves the best performance.

**Time-period sampling lengths within score distillation.** We demonstrated the effectiveness of the high-frequency enhanced score distillation technique and the time-aware discriminator within the proposed time-aware distillation framework in Sec. A.3.1. In this section, we further investigate the impact of sampling time steps on model performance within the high-frequency enhanced score distillation technique. Specifically, we divide the total time steps into five equal periods and incrementally increase the number of sampled periods to assess model performance on RealSR and RealSet65 datasets. As shown in Table 10, the highest CLIPIQA and MUSIQ scores were achieved by calculating the score distillation loss during small time steps. Since the diffusion model primarily focuses on high-frequency details during small time steps, this result corroborates our analysis in Sec. 3.1. In comparison to the teacher model, the student model exhibits a notable deficiency in modeling high-frequency details, making it both reasonable and effective to compute the score distillation loss at small time steps.

### A.3.2 EXPERIMENTAL RESULTS ON SD-BASED SR METHOD

In addition to distilling the super-resolution model trained from scratch, we also apply our proposed TAD-SR to distill the SOTA SD-based super-resolution model to further validate its effectiveness.

**Training Datasets**. We adopt DIV2K (Agustsson & Timofte, 2017), Flickr2K (Timofte et al., 2017), first 20K images from LSDIR (Li et al., 2023) and first 10K face images from FFHQ (Karras et al., 2019) for training. The degradation pipeline of Real-ESRGAN (Wang et al., 2021b) is used to synthesize LR-HR training pairs.

Table 12: Quantitative comparison with state of the arts on RealLR200 dataset dataset. The best and second best results are highlighted in **bold** and underline. Note that since the RealLR200 dataset lacks high-resolution images, we only computed non-reference metrics.

| Methods | RealLR200 | | | |
|---|---|---|---|---|
| | NIQE↓ | CLIPIQA↑ | MUSIQ↑ | MANIQA↑ |
| BSRGAN | 4.38 | 0.570 | 64.87 | 0.369 |
| RealESRGAN | 4.20 | 0.542 | 62.93 | 0.366 |
| LDL | 4.38 | 0.509 | 60.95 | 0.327 |
| FeMaSR | 4.34 | 0.655 | 64.24 | 0.410 |
| StableSR-200 | 4.25 | 0.592 | 62.89 | 0.367 |
| ResShift-15 | 6.29 | 0.647 | 60.25 | 0.418 |
| PASD-20 | 4.18 | 0.620 | 66.35 | 0.419 |
| SeeSR-50 | 4.16 | 0.662 | 68.63 | **0.491** |
| +UniPC-10 | 4.25 | 0.601 | 66.90 | 0.433 |
| +DPMSolver-10 | 4.28 | 0.603 | 66.92 | 0.435 |
| SinSR-1 | 5.62 | **0.697** | 63.85 | 0.445 |
| AddSR-1 | 4.06 | 0.585 | 66.86 | 0.418 |
| OSEDiff-1 | 4.05 | 0.674 | **69.61** | 0.444 |
| TAD-SR-1 | **3.95** | 0.674 | 69.48 | 0.482 |

**Testing Datasets**. We evaluate TAD-SR on two real-world datasets: RealSR (Cai et al., 2019) and RealLR200 (Wu et al., 2024b), as well as one one synthetic dataset, DIV2K-val(Agustsson & Timofte, 2017). The method for acquiring HR-LR image pairs in the DIV2K dataset follows the procedure detailed in (Wang et al., 2023b), and except RealLR200, all datasets are cropped to 512×512 patches.

**Compared Methods.** We compare our SeeSR with several state-of-the-art Real-ISR methods, which can be categorized into two groups. The first group consists of GAN-based methods, including BSRGAN (Zhang et al., 2021), Real-ESRGAN (Karras et al., 2019), LDL (Liang et al., 2022a), FeMaSR (Chen et al., 2022). The second group consists of recent diffusion-based methods, including StableSR (Wang et al., 2023b), ResShift (Yue et al., 2024), PASD (Yang et al., 2023), SeeSR (Wu et al., 2024b), SinSR (Wang et al., 2023c), AddSR (Xie et al., 2024) and OSEDiff (Wu et al., 2024a). Additionally, we applied samplers such as UniPC (Zhao et al., 2024) and DPM-Solver (Lu et al., 2022) to the inference process of the teacher model SeeSR and used them as baselines.

**Evaluation Metrics.** We employ non-reference metrics (e.g., MANIQA (Yang et al., 2022), MUSIQ (Ke et al., 2021), CLIPIQA (Wang et al., 2023a) and NIQE (Zhang et al., 2015)) and reference metrics (e.g., LPIPS (Zhang et al., 2018a), PSNR and FID (Heusel et al., 2017)) to comprehensively evaluate our TAD-SR. Note that in real-world super-resolution tasks, the non-reference metrics are more aligned with human perception and better reflects the subjective quality of images.

**Evaluation results.** We first show the quantitative comparison on one synthetic dataset and two real-world datasets in Tables 11, 12 and 13. The observations from the table are as follows: (1) The GAN-based method shows advantages over diffusion-based methods in full-reference metrics (*e.g.,* PSNR and LPIPS), yet it significantly lags behind diffusion-based methods in non-reference metrics. (2) Our method achieves performance comparable to the teacher model (SeeSR) using only single-step sampling. (3) Compared to other one-step diffusion-based SR methods, our approach outperforms in most metrics. Furthermore, unlike the concurrent work OSEDiff (Wu et al., 2024a), our method is more versatile, allowing it to accelerate any diffusion-based SR models for practical needs. Additionally, the visualization results demonstrate that our method not only enhances image details with greater clarity (as illustrated in the second row of Fig. 8) but also preserves the similarity to the original image as much as possible (as shown in the fourth row of Fig. 8). Additionally, we also report the inference time of different SD-based SR methods as shown in Table 14.Overall, our TAD-SR can effectively and efficiently complete image super-resolution reconstruction.

Table 13: Quantitative comparison with state of the arts on DIV2k-val dataset. The best and second best results are highlighted in **bold** and underline.

| Methods | DIV2K-val | | | | | | |
|---|---|---|---|---|---|---|---|
| | PSNR↑ | LPIPS ↓ | FID↓ | NIQE↓ | CLIPIQA↑ | MUSIQ↑ | MANIQA↑ |
| BSRGAN | 24.58 | 0.335 | 44.22 | 4.75 | 0.524 | 61.19 | 0.356 |
| RealESRGAN | 24.29 | 0.311 | 37.64 | 4.68 | 0.527 | 61.06 | 0.382 |
| LDL | 23.83 | 0.326 | 42.28 | 4.86 | 0.518 | 60.04 | 0.375 |
| FeMaSR | 23.06 | 0.346 | 53.70 | 4.74 | 0.599 | 60.82 | 0.346 |
| StableSR-200 | 23.29 | 0.312 | **24.54** | 4.75 | 0.676 | 65.83 | 0.422 |
| ResShift-15 | **24.72** | 0.340 | 41.99 | 6.47 | 0.594 | 60.89 | 0.399 |
| PASD-20 | 24.51 | 0.392 | 31.58 | 5.37 | 0.551 | 59.99 | 0.399 |
| SeeSR-50 | 23.68 | 0.319 | 25.97 | 4.81 | **0.693** | **68.68** | **0.504** |
| +UniPC-10 | 24.07 | 0.339 | 27.33 | 5.00 | 0.607 | 64.97 | 0.432 |
| +DPMSolver-10 | 24.12 | 0.338 | 27.32 | 5.03 | 0.612 | 65.07 | 0.435 |
| SinSR-1 | 24.41 | 0.324 | 35.23 | 6.01 | 0.648 | 62.80 | 0.424 |
| AddSR-1 | 23.26 | 0.362 | 29.68 | 4.76 | 0.573 | 63.69 | 0.405 |
| OSEDiff-1 | 23.72 | **0.294** | 26.33 | 4.71 | 0.661 | 67.96 | 0.443 |
| TAD-SR-1 | 23.54 | 0.311 | 25.96 | **4.64** | 0.664 | 67.01 | 0.470 |

Table 14: Complexity comparison among different SD-based SR methods. All methods are tested on the ×4 (128→512) SR tasks, and the inference time is measured on an V100 GPU.

| Method | StableSR | PASD | SeeSR | AddSR | OSEDiff | TAD-SR |
|---|---|---|---|---|---|---|
| NFE | 200 | 20 | 50 | 1 | 1 | 1 |
| Inference time (s) | 17.76 | 13.51 | 8.40 | 0.64 | 0.48 | 0.64 |

## A.4 LIMITATIONS

Although our TAD-SR demonstrates strong performance, it shares a common limitation with current single-step distillation methods: increasing the number of inference steps alone does not yield better performance. Thus, developing a distillation method that matches the performance of state-of-the-art single-step approaches while enabling additional inference steps to enhance performance is a key area of our ongoing research.

## A.5 MORE VISUALIZATION RESULTS

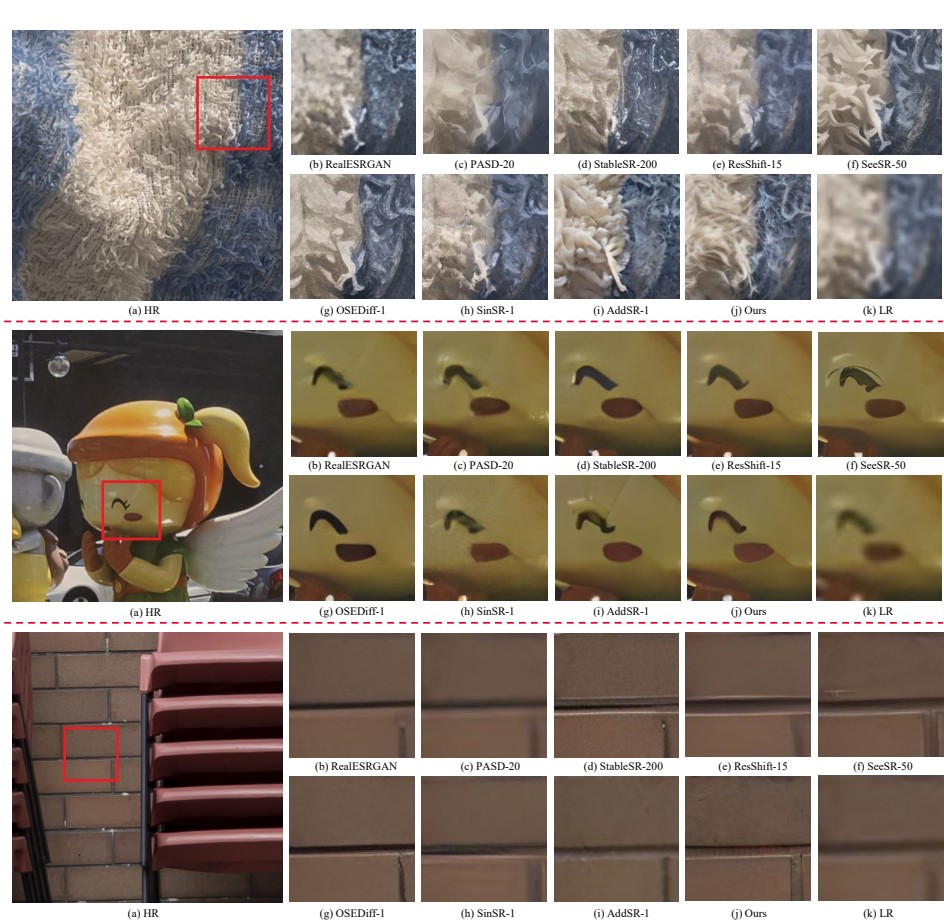

Figure 8: Visual comparison on real-world LR images. Note that SeeSR is the teacher model.

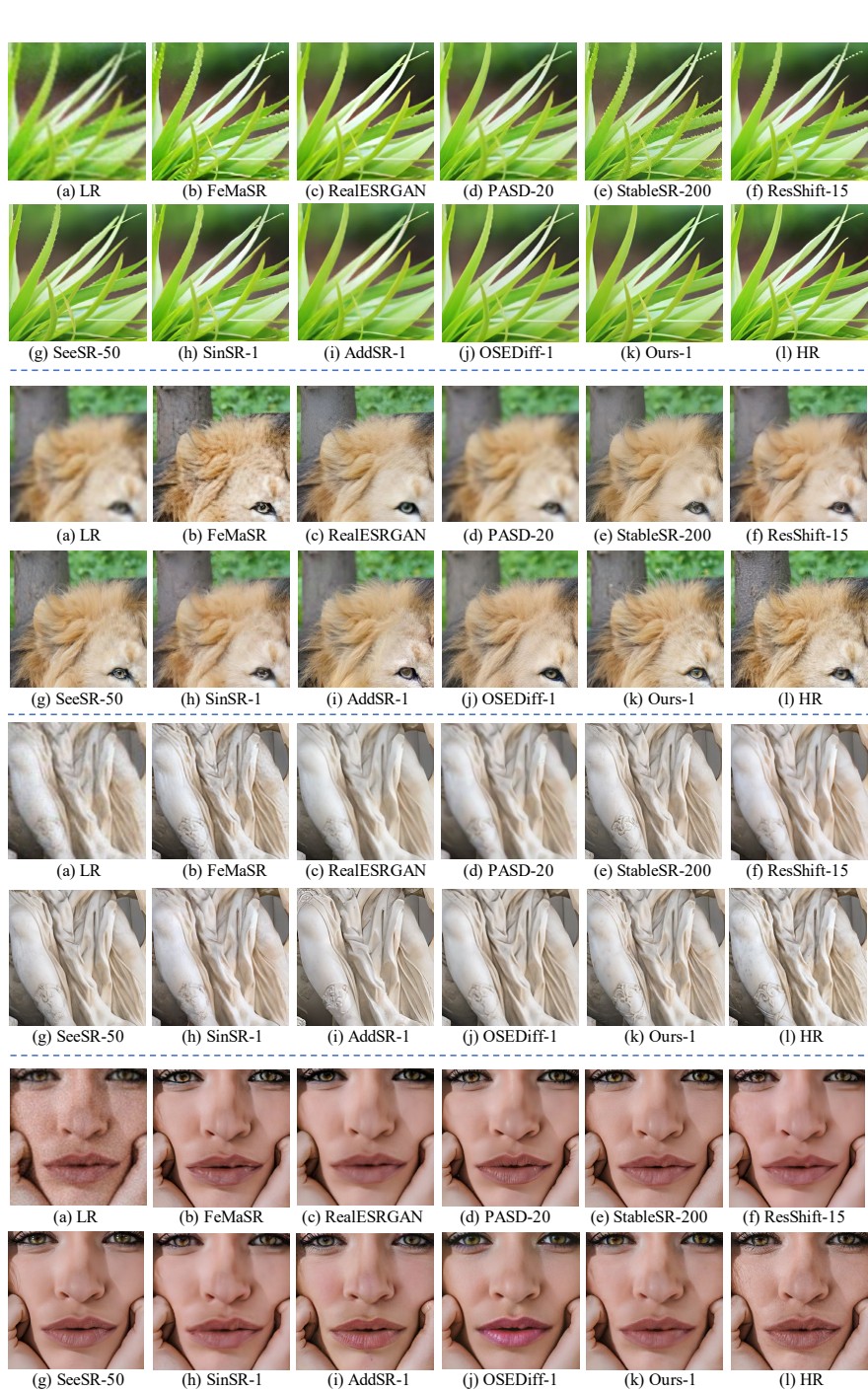

Figure 9: Qualitative comparisons of different methods on four synthetic examples of the *DIV2K* dataset. SeeSR is the teacher model.

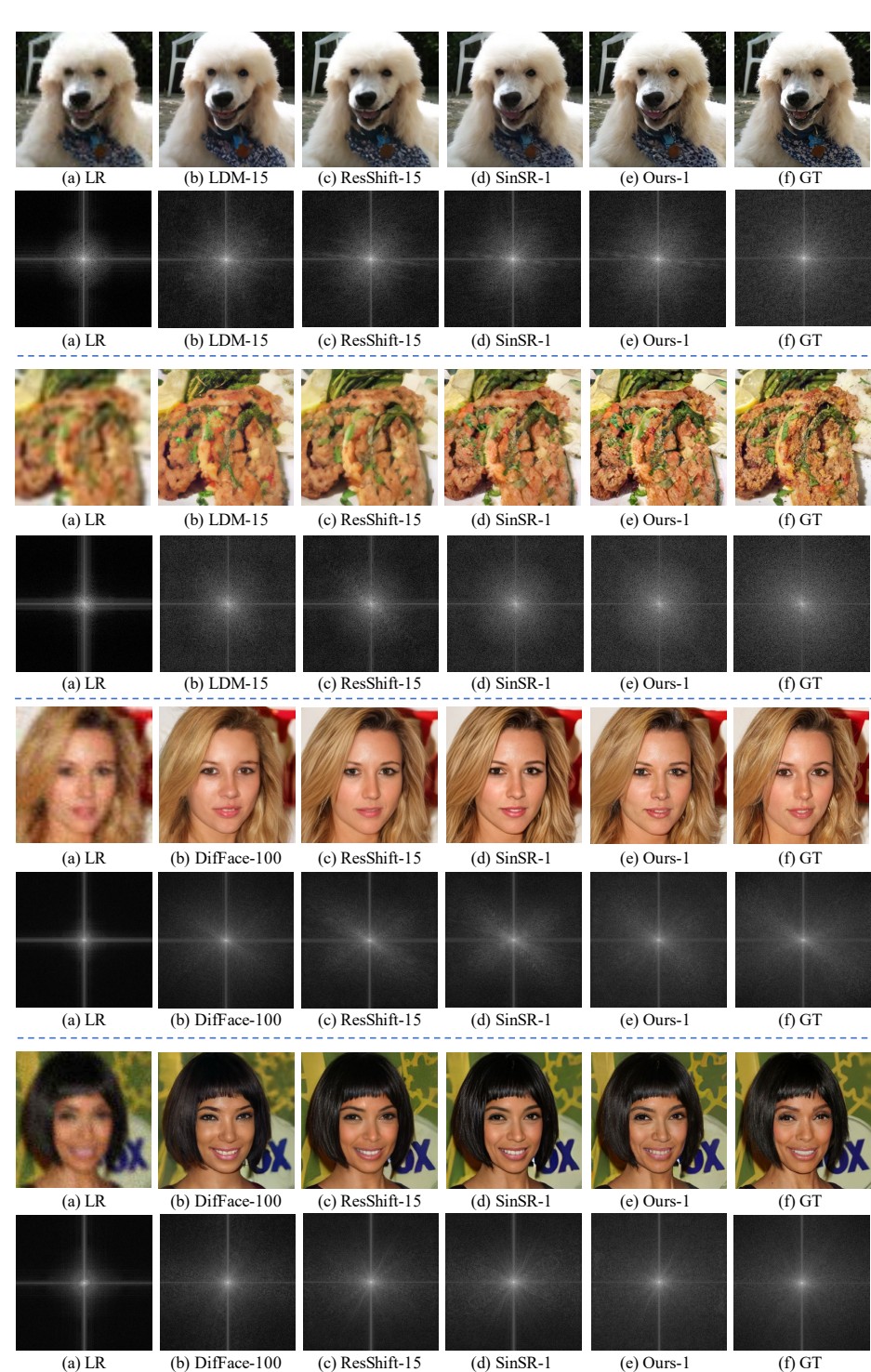

Figure 10: The visualizations of images generated by different SR methods, along with their Fourier-transformed spectrograms, reveal that our method preserves more high-frequency information than other methods. Please zoom in for a better view.

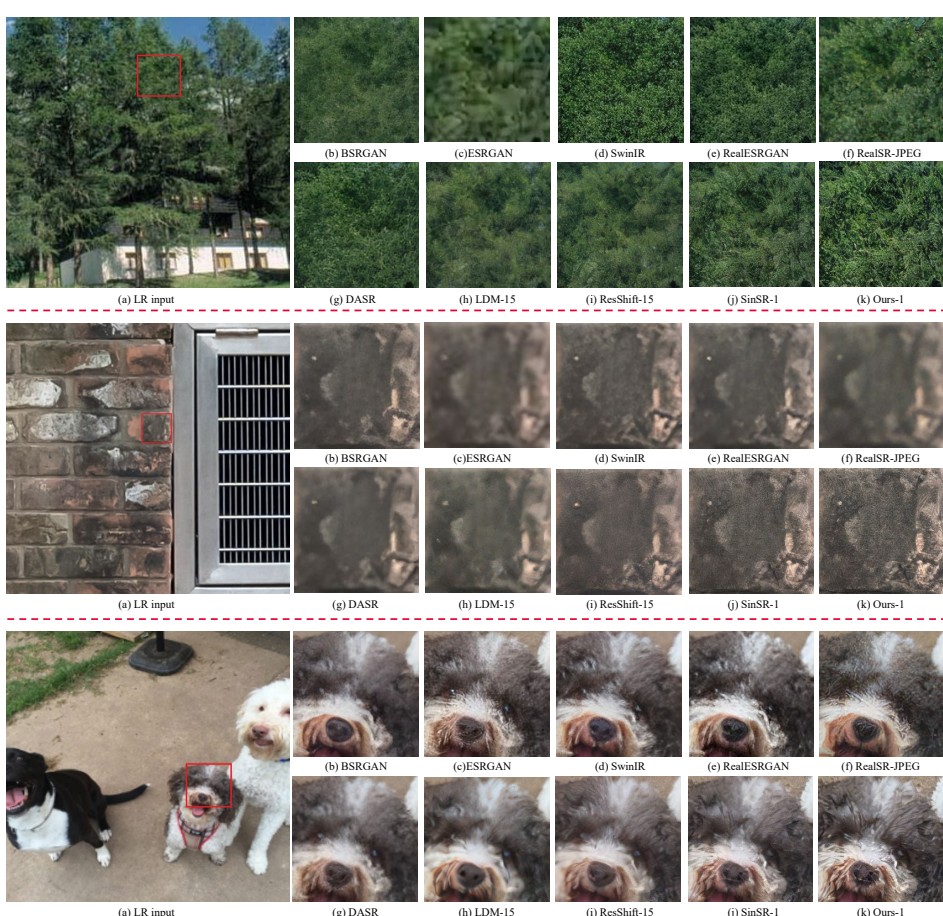

Figure 11: Qualitative comparisons of different methods on three real-world examples of the *RealSR* and *RealSet65* dataset. Please zoom in for a better view.

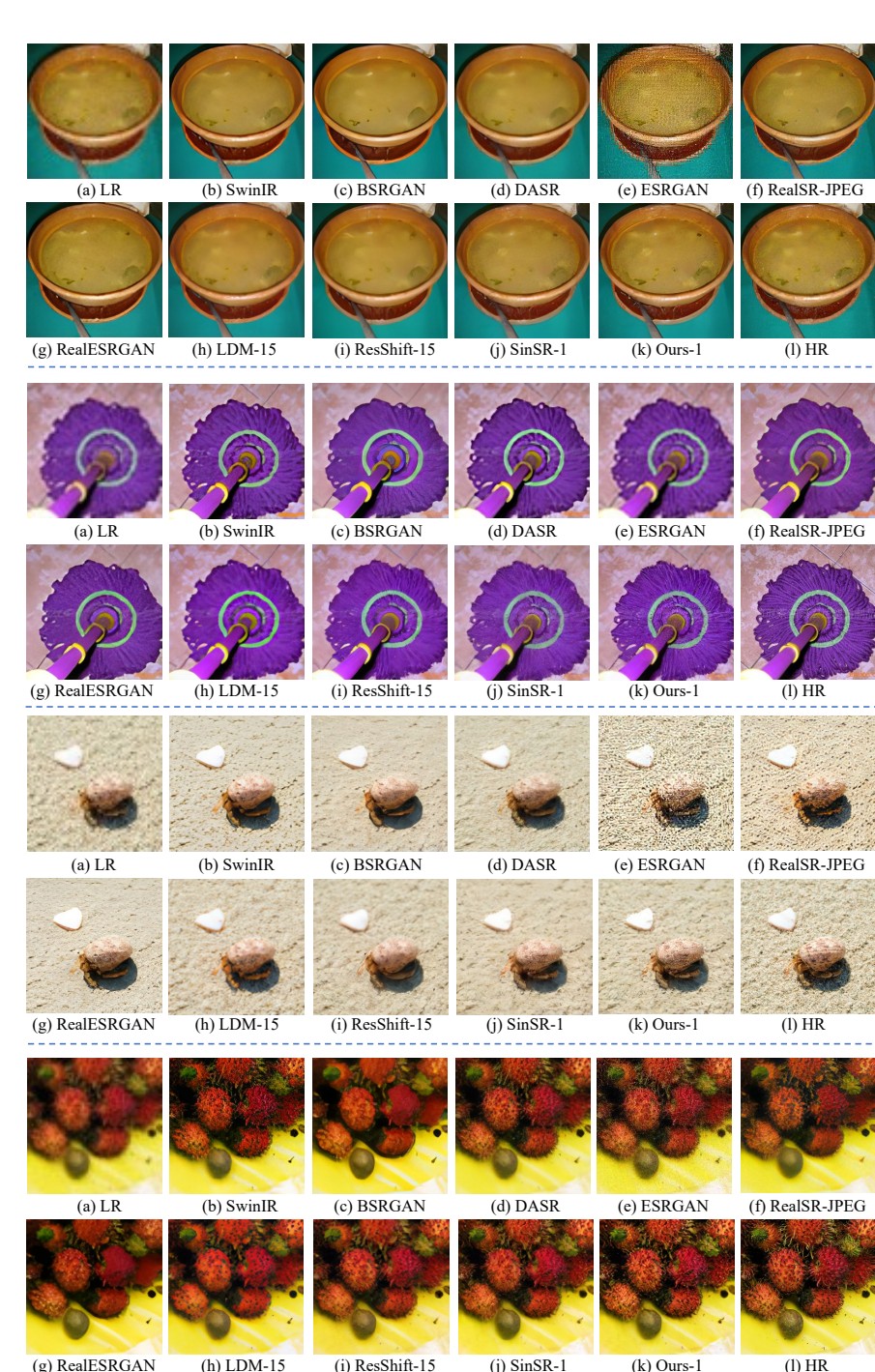

Figure 12: Qualitative comparisons of different methods on four synthetic examples of the *ImageNet-Test* dataset.

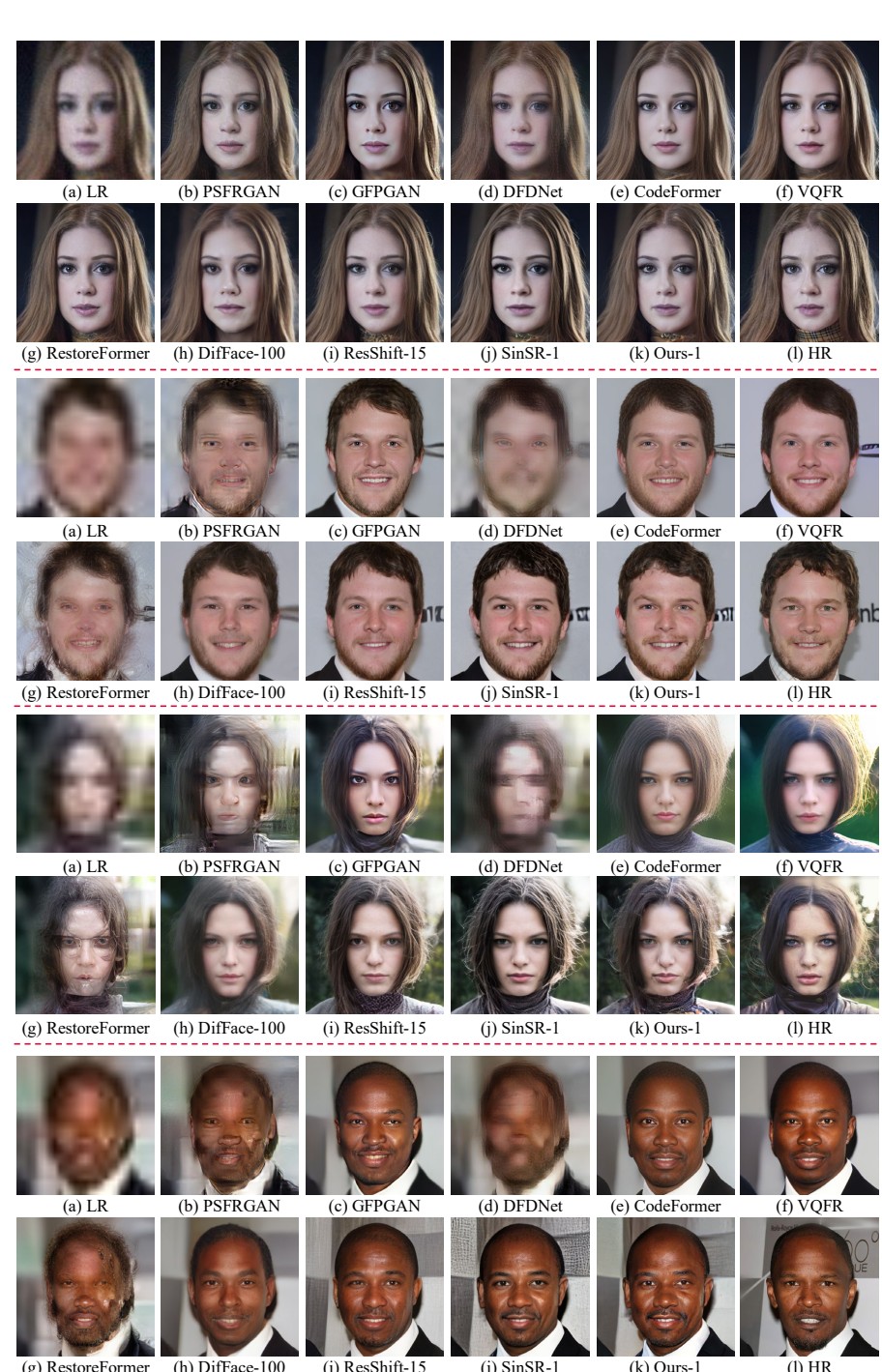

Figure 13: Qualitative comparisons of different methods on four synthetic examples of the *CelebA-Test* dataset.

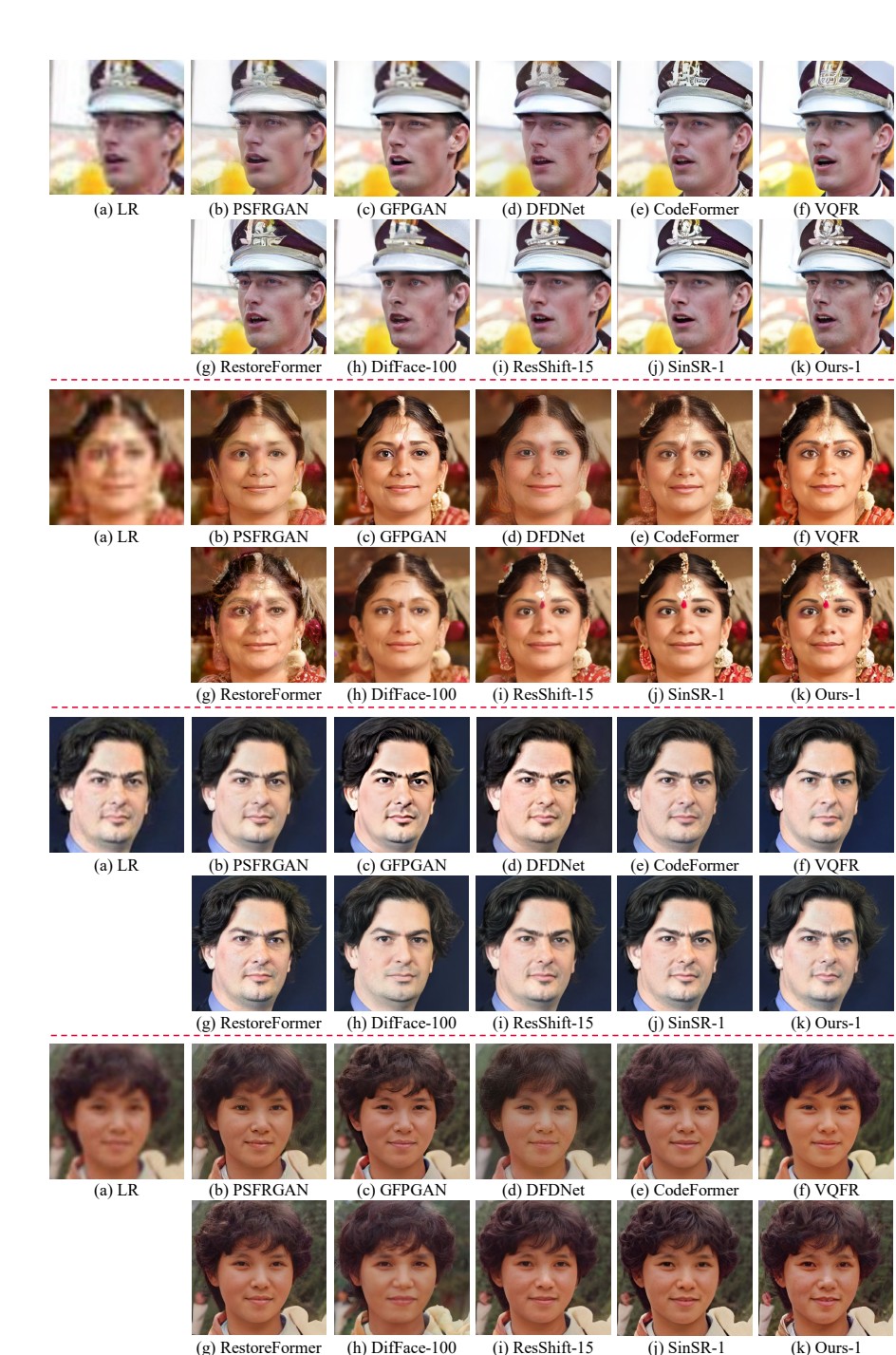

Figure 14: Qualitative comparisons of different methods on four real-world examples of the *LFW*, *WebPhoto* and *WIDER* dataset.

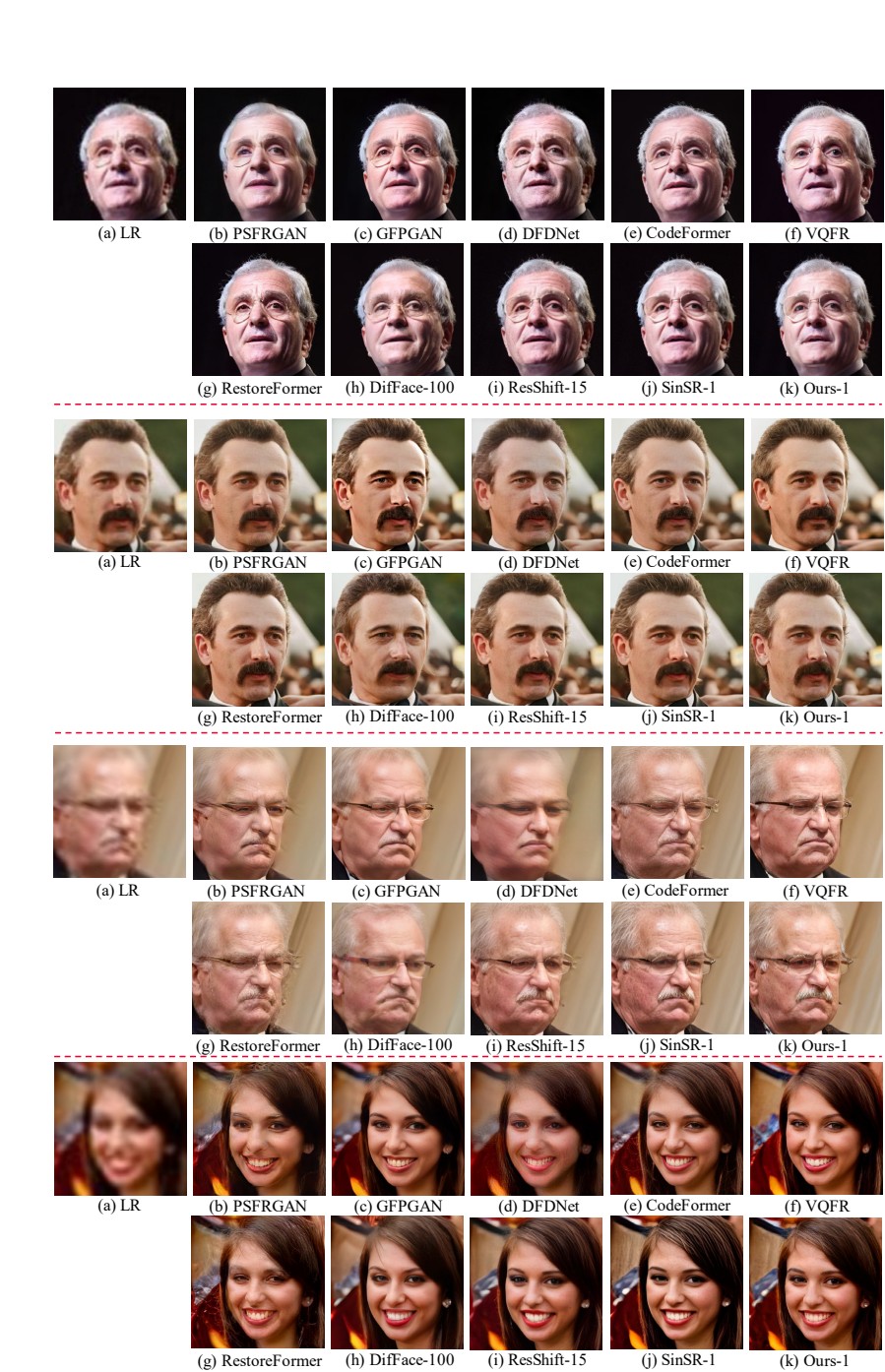

Figure 15: Qualitative comparisons of different methods on four real-world examples of the *LFW*, *WebPhoto* and *WIDER* dataset.

