# OpenReview forum: "One Step Diffusion-based Super-Resolution with Time-Aware Distillation"
_ICLR.cc/2025/Conference — Submitted to ICLR 2025_

### Official Review · Reviewer_Rnto · 2024-11-02

**Soundness:** 2
**Presentation:** 3
**Contribution:** 2
**Rating:** 6
**Confidence:** 4

**Summary:**

The author proposes a time-aware diffusion distillation method, named TAD-SR, where a novel score distillation strategy is introduced to align the score functions between the outputs of the student and teacher models after minor noise perturbation. Such distillation strategy eliminates the inherent bias in score distillation sampling (SDS) and enables the student models to focus more on high-frequency image details by sampling at smaller time steps.  Furthermore, a time-aware discriminator is designed to mitigate performance limitations stemming from distillation, which distinguishes the diffused distributions of real and generated images under varying noise disturbance levels by injecting time information.

**Strengths:**

1. The proposed distillation strategy is simple and straightforward, which can eliminate the inherent bias in score distillation sampling (SDS) and enable the student models to focus more on high-frequency image details.
2. The proposed time-aware discriminator can differentiate between real and synthetic data, contributing to the generation of high-quality images.
3. The presentation of this work is written well and is easy to read.

**Weaknesses:**

1. It is confusing which is the final output of the model when inference, z_0^{stu} or z ̂_0^{stu}? It is not clearly indicated in Figure 4. Please explicitly state in the text and figure.
2. The authors should clarify if the teacher model is used at all during inference, or if it is only used during training. If I understand correctly, only the student model samples one step, and then the teacher model is used later to sample multiple steps to get the final clean latent, so the model performance relies heavily on the performance of the teacher model, and is not exactly efficient.
3. What is the purpose of setting the weighting function (ω = 1/CS )? Please provide intuition for why this weighting function was chosen, and what effect it has on the training process or results.
4. In order to eliminate the dependence of the proposed method on the teacher model of ResShift, the relevant ablation experiments should be conducted by replacing the different teacher models to validate the effectiveness of the proposed method.
5. The experiments lack comparisons with the most relevant distillation methods, including DMD, DEQ[1], DFOSD[2], etc. Among them, DMD, a new diffusion model, utilizes similar score distillation techniques to the proposed HSD. DEQ and DFOSD are both efficient and relevant diffusion models, which require one-step diffusion distillation or even no distillation.
6. In the experimental section, the authors compare many GAN and transformer-related methods. However, the proposed method is a diffusion model and should be compared with the most relevant diffusion models to validate its efficiency, especially accelerated diffusion models, including OSEDiff[3], DPM++[4], Unipc[5], etc.
7. The authors claim that the method is designed to accomplish effective and efficient image super-resolution, but did not include a complexity comparison of the different methods (including parameters, sampling steps, running time, MACs, etc.), which is crucial for diffusion models. Please provide a Table to compare these computational complexity metrics with the key baselines.
8. Are there any limit conditions for using the method? The author should discuss and analyze the limitations of the proposed method. It is recommended to add a discussion of a discussion of potential limitations or where the proposed method might not perform as well.

References

[1] Geng Z, Pokle A, Kolter J Z. One-step diffusion distillation via deep equilibrium models[C]. Advances in Neural Information Processing Systems, 2024.

[2] Li J, Cao J, Zou Z, et al. Distillation-free one-dtep diffusion for real-world image super-resolution[J]. arxiv preprint arxiv:2410.04224, 2024.

[3] Wu R, Sun L, Ma Z, et al. One-step effective diffusion network for real-world image super-resolution[J]. arxiv preprint arxiv:2406.08177, 2024.

[4] Lu C, Zhou Y, Bao F, et al. Dpm-solver++: Fast solver for guided sampling of diffusion probabilistic models[J]. arxiv preprint arxiv:2211.01095, 2022.

[5] Zhao W, Bai L, Rao Y, et al. Unipc: A unified predictor-corrector framework for fast sampling of diffusion models[C]. Advances in Neural Information Processing Systems, 2024.

**Questions:**

See the Weakness part.
The author should carefully describe the details of the method to enhance the readability and clarity of the paper. In addition, the comparison of the most relevant methods (including complexity comparison) should be added to clarify the innovation and effectiveness of the method, and the advancement of the method should be proved through relevant experiments.

I tend to improve the score if the author can solve my concerns.

---

> ### Author Response · Authors · 2024-11-20
> **Response to Reviewer Rnto**
>
> Thank you for your comments and feedback. We address your concerns here.
>
> >**Q1**: It is confusing which is the final output of the model when inference, z_0^{stu} or z ̂_0^{stu}? It is not clearly indicated in Figure 4. Please explicitly state in the text and figure.
>
> >**A1**: Thank you for pointing out this issue. $z_0^{stu}$ is the final output of the student model. $\hat{z}_0^{stu}$ represents the clean value predicted by the teacher model after re-adding noise to the output of the student model. This value is used to calculate the loss. We will revise the paper and the images in the manuscript to enhance clarity and make them easier to understand.
>
> >**Q2**: The authors should clarify if the teacher model is used at all during inference, or if it is only used during training. If I understand correctly, only the student model samples one step, and then the teacher model is used later to sample multiple steps to get the final clean latent, so the model performance relies heavily on the performance of the teacher model, and is not exactly efficient.
>
> >**A2**: Thank you for your suggestion. As the reviewer understands, the teacher model is only used during the training. Additionally, we not only leveraged the knowledge from the teacher model but also incorporated the ground truth (GT) into the distillation framework through adversarial learning to provide additional supervision for the model. Therefore, the performance of our method is not solely dependent on the teacher model’s performance.
>
> >**Q3**: What is the purpose of setting the weighting function (ω = 1/CS )? Please provide intuition for why this weighting function was chosen, and what effect it has on the training process or results.
>
> >**A3**: Apologies for the confusion. What we intended to convey is that our score distillation loss is averaged over both spatial and channel dimensions, which facilitates model optimization [1][2]. However, there was an error in the formula expression, and we will correct this in the next version of the manuscript.
>
> >**Q4**: In order to eliminate the dependence of the proposed method on the teacher model of ResShift, the relevant ablation experiments should be conducted by replacing the different teacher models to validate the effectiveness of the proposed method.
>
> >**A4**: Thank you for your suggestion. We have included the results of distilling the SD-based SR method SeeSR into a single step using TAD-SR. The quantitative and qualitative experimental results are presented in Tables 1, 2, and 3. As shown in the tables, our proposed distillation method demonstrates strong generalization capabilities, effectively distilling different teacher models into a single step and generating promising results.
>
> Table 1: Quantitative comparison with state of the arts on RealSR dataset. Following the experimental setup of SeeSR, the LR images in the RealSR dataset were center-cropped to 128 $\times$ 128. The best and second best results are highlighted in bold and italic.
> |        Methods       |    PSNR  $\uparrow$  |   LPIPS $\downarrow$  |     FID  $\downarrow$  |   NIQE  $\downarrow$ |  CLIPIQA $\uparrow$  |   MUSIQ  $\uparrow$  |  MANIQA $\uparrow$  |
> |--------------------|:---------:|:---------:|:----------:|:--------:|:---------:|:---------:|:--------:|
> |        BSRGAN        |  _26.49_  | **0.267** |   141.28   |   5.66   |   0.512   |   63.28   |   0.376  |
> |      RealESRGAN      |   25.78   |  _0.273_  |   135.18   |   5.83   |   0.449   |   60.36   |   0.373  |
> |          LDL         |   25.09   |   0.277   |   142.71   |     6.00    |    0.430   |   58.04   |   0.342  |
> |        FeMaSR        |   25.17   |   0.294   |   141.05   |   5.79   |   0.541   |   59.06   |   0.361  |
> |     StableSR-200     |   25.63   |   0.302   |    133.40   |   5.76  |   0.528   |   61.11   |   0.366  |
> |      ResShift-15     |   26.34   |   0.346   |   149.54   |   6.87  |   0.542   |   56.06   |   0.375  |
> |        PASD-20       | **26.67** |   0.344   |   _122.30_  |   6.06  |   0.519   |   62.92   |   0.404  |
> |       SeeSR-50       |   25.24   |   0.301   |   125.42   |  _5.39_  |   _0.670_  | **69.82** | **0.540** |
> |    SeeSR(UniPC-10)   |   25.86   |   0.281   |   122.41   |   5.53  |   0.577   |   67.12   |   0.476  |
> | SeeSR(DPMSolver-10) |    25.90   |   0.281   |   122.46   |   5.54  |   0.581   |   67.12   |   0.478  |
> |        SinSR-1       |   26.16   |   0.308   |   142.44   |   5.75   |    0.630   |   60.96   |   0.399  |
> |        AddSR-1       |   23.12   |   0.309   |   132.01   |   5.54   |   0.552   |   67.14   |   0.488  |
> |       OSEDiff-1      |   25.15   |   0.292   |   123.49   |   5.63   |   0.668   |   68.99   |   0.474  |
> |       TAD-SR-1       |    24.50   |   0.304   | **118.38** | **5.13** | **0.676** |  _69.02_  |  _0.526_ |

---

> > ### Author Response · Authors · 2024-11-20
> > **Response to Reviewer Rnto**
> >
> > Table 2: Quantitative comparison with state of the arts on RealLR200 dataset dataset. The best and second best results are highlighted in bold and italic. Note that since the RealLR200 dataset lacks high-resolution images, we only computed non-reference metrics.
> >
> > |        Methods       |   NIQE $\downarrow$  |  CLIPIQA $\uparrow$ |   MUSIQ  $\uparrow$ |   MANIQA $\uparrow$ |
> > |--------------------|:--------:|:---------:|:---------:|:---------:|
> > |        BSRGAN        |   4.38   |   0.570   |   64.87   |   0.369   |
> > |      RealESRGAN      |   4.20   |   0.542   |   62.93   |   0.366   |
> > |          LDL         |   4.38   |   0.509   |   60.95   |   0.327   |
> > |        FeMaSR        |   4.34   |   0.655   |   64.24   |   0.410   |
> > |     StableSR-200     |   4.25   |   0.592   |   62.89   |   0.367   |
> > |      ResShift-15     |   6.29   |   0.647   |   60.25   |   0.418   |
> > |        PASD-20       |   4.18   |   0.620   |   66.35   |   0.419   |
> > |       SeeSR-50       |   4.16   |   0.662   |   68.63   | **0.491** |
> > |    SeeSR(UniPC-10)   |   4.25   |   0.601   |    66.90   |   0.433   |
> > | SeeSR(DPMSolver-10) |   4.28   |   0.603   |   66.92   |   0.435   |
> > |        SinSR-1       |   5.62   | **0.697** |   63.85   |   0.445   |
> > |        AddSR-1       |   4.06   |   0.585   |   66.86   |   0.418   |
> > |       OSEDiff-1      |  _4.05_  |  _0.674_  | **69.61** |   0.444   |
> > |       TAD-SR-1       | **3.95** |  _0.674_  |  _69.48_  |  _0.482_  |
> >
> > Table 3: Quantitative comparison with state of the arts on DIV2k-val dataset. The best and second best results are highlighted in bold and italic.
> >
> > |        Methods       |    PSNR $\uparrow$ |   LPIPS $\downarrow$  |    FID $\downarrow$   |   NIQE  $\downarrow$ |  CLIPIQA $\uparrow$ |   MUSIQ  $\uparrow$ |   MANIQA $\uparrow$ |
> > |--------------------|:---------:|:---------:|:---------:|:--------:|:---------:|:---------:|:---------:|
> > |        BSRGAN        |  _24.58_  |   0.335   |   44.22   |   4.75   |   0.524   |   61.19   |   0.356   |
> > |      RealESRGAN      |   24.29   |  _0.311_  |   37.64   |   4.68   |   0.527   |   61.06   |   0.382   |
> > |          LDL         |   23.83   |   0.326   |   42.28   |   4.86   |   0.518   |   60.04   |   0.375   |
> > |        FeMaSR        |   23.06   |   0.346   |   53.70   |   4.74   |   0.599   |   60.82   |   0.346   |
> > |     StableSR-200     |   23.29   |   0.312   | **24.54** |   4.75   |  _0.676_  |   65.83   |   0.422   |
> > |      ResShift-15     | **24.72** |    0.34   |   41.99   |   6.47   |   0.594   |   60.89   |   0.399   |
> > |        PASD-20       |   24.51   |   0.392   |   31.58   |   5.37   |   0.551   |   59.99   |   0.399   |
> > |       SeeSR-50       |   23.68   |   0.319   |   25.97   |   4.81   | **0.693** | **68.68** | **0.504** |
> > |    SeeSR(UniPC-10)   |   24.07   |   0.339   |   27.33   |   5.00   |   0.607   |   64.97   |   0.432   |
> > | SeeSR(DPMSolver-10) |   24.12   |   0.338   |   27.32   |   5.03   |   0.612   |   65.07   |   0.435   |
> > |        SinSR-1       |   24.41   |   0.324   |   35.23   |   6.01   |   0.648   |   62.80   |   0.424   |
> > |        AddSR-1       |   23.26   |   0.362   |   29.68   |   4.76   |   0.573   |   63.69   |   0.405   |
> > |       OSEDiff-1      |   23.72   | **0.294** |   26.33   |  _4.71_  |   0.661   |  _67.96_  |   0.443   |
> > |       TAD-SR-1       |   23.54   |  _0.311_  |  _25.96_  | **4.64** |   0.664   |   67.01   |  _0.470_  |
> >
> > >**Q5**: The experiments lack comparisons with the most relevant distillation methods, including DMD, DEQ[1], DFOSD[2], etc. Among them, DMD, a new diffusion model, utilizes similar score distillation techniques to the proposed HSD. DEQ and DFOSD are both efficient and relevant diffusion models, which require one-step diffusion distillation or even no distillation.
> >
> > >**A5**: Thank you for your suggestion. We applied DMD to super-resolution tasks and compared it with our proposed method. From Table 4, it can be seen that while DMD achieves promising results when transferred to super-resolution tasks, it remains inferior to our approach. Regarding DEQ[1], its high training cost makes applying it to super-resolution tasks extremely challenging. As noted in its original paper, DEQ experiments were only conducted on the CIFAR-10 dataset due to these limitations. For DFOSD[2], we found that its code is not open source, and the training relied on a self-collected dataset that is not publicly available, making it difficult to perform a fair comparison with our method.
> >
> > >To further validate the effectiveness of our approach, we applied TAD-SR to unconditional generation tasks and compared it with DMD and DEQ on the CIFAR-10 dataset. The experimental results are presented in Table 5. The results demonstrate that our method performs well in unconditional generation tasks, surpassing both DMD and DEQ.

---

> ### Author Response · Authors · 2024-11-20
> **Response to Reviewer Rnto**
>
> Table 4: Quantitative results of different SR methods. The best and second best results are highlighted in bold and italic. ∗ indicates that the result was obtained by replicating the method in the paper.
> |   Datasets  |   |  ImageNet-test |    |  RealSR  |    RealSR  |  RealSet65  |    RealSet65  |
> |-----------|:---------:|:---------:|:----------:|:---------:|:----------:|:---------:|:----------:|
> |   Methods   |  LPIPS  $\downarrow$ |  CLIPIQA $\uparrow$ |    MUSIQ $\uparrow$  |  CLIPIQA $\uparrow$   |    MUSIQ $\uparrow$   |  CLIPIQA $\uparrow$  |    MUSIQ  $\uparrow$  |
> |    LDM-15   |   0.269   |   0.512   |   46.419   |   0.384   |   49.317   |   0.427   |   47.488   |
> | ResShift-15 |   0.231   |   0.592   |    53.660  |  0.596   |   59.873   |   0.654   |   61.330   |
> |   SinSR-1   | **0.221** |   0.611   |   53.357   | 0.689   |   61.582   |   0.715   |   62.169   |
> |   SinSR*-1  |   0.231   |   0.599   |   52.462   |  0.691   |   60.865   |   0.712   |   62.575   |
> |    DMD*-1   |   0.246   |  _0.612_  |   54.124   | _0.709_  |   _63.610_  |  _0.723_  |  _66.177_  |
> |   TAD-SR-1  |  _0.227_  | **0.652** | **57.533** |  **0.741** | **65.701** | **0.734** | **67.500** |
>
> Table 5: Generative performance on unconditional CIFAR-10. The best results are highlighted in bold.
> | Method | DDPM | DDIM | EDM(Teacher) | DPM-solver2 | UniPC | CD-L2 | CD-LPIPS |  DEQ |  DMD |  Ours  |
> |:------:|:----:|:----:|:------------:|:-----------:|:-----:|:-----:|:--------:|:----:|:----:|:------:|
> |   NFE $\downarrow$ | 1000 |  50  |      35      |      12     |   8   |   1   |     1    |   1  |   1  |    1   |
> |   FID $\downarrow$ | 3.17 | 4.67 |   **1.88**   |     5.28    |  5.10  |  7.90  |   3.55   | 6.91 | 3.77 | _2.31_ |
>
> >**Q6**: In the experimental section, the authors compare many GAN and transformer-related methods. However, the proposed method is a diffusion model and should be compared with the most relevant diffusion models to validate its efficiency, especially accelerated diffusion models, including OSEDiff[3], DPM++[4], Unipc[5], etc.
>
> >**A6**: Thank you for your suggestion. Since OSEDiff is an SD-based SR method, we compared our approach to OSEDiff while distilling the SD-based SR model SeeSR. This ensures a fair comparison, as both methods were trained on the same dataset. As shown in Tables 1, 2, and 3, our method outperforms OSEDiff across most evaluation metrics.
>
> >In response to the reviewers' suggestions, we have also incorporated the designed sampler methods, Unipc[5] and DPM++[4], into Tables 1, 2, and 3. (Note that we did not apply these samplers to ResShift, as ResShift modifies the standard Markov chain, creating challenges for its adaptation to these samplers.) Despite this, the results clearly demonstrate that our method significantly outperforms methods employing these samplers.
>
> >**Q7**: The authors claim that the method is designed to accomplish effective and efficient image super-resolution, but did not include a complexity comparison of the different methods (including parameters, sampling steps, running time, MACs, etc.), which is crucial for diffusion models. Please provide a Table to compare these computational complexity metrics with the key baselines.
>
> >**A7**: Based on the reviewers' feedback, we have included a complexity comparison between TAD-SR and baseline methods, as presented in Tables 6 and 7. Table 6 focuses on comparisons with GAN-based methods and diffusion-based super-resolution methods trained from scratch. The results demonstrate that TAD-SR accelerates the teacher model, ResShift, to a single inference step, improving its speed by approximately tenfold. Table 7 highlights a comparison of inference time with SD-based super-resolution methods, revealing that our method's inference delay is only 7.6% of the teacher model, SeeSR.
>
> Table 6: Complexity comparison among different SR methods. All methods are tested on the ×4 (64→256) SR tasks, and the inference time is measured on an A100 GPU.
> |  Method | ESRGAN | RealSR-JPEG | BSRGAN | SwinIR | RealESRGAN |  DASR |  LDM  | ResShift | SinSR | TAD-SR |
> |-------|:------:|:-----------:|:------:|:------:|:----------:|:-----:|:-----:|:--------:|:-----:|:------:|
> |   NFE   |    1   |      1      |    1   |    1   |      1     |   1   |   15  |    15    |   1   |    1   |
> | Inference time（s） |  0.038 |    0.038    |  0.038 |  0.107 |    0.038   | 0.022 | 0.408 |   0.682  | 0.058 |  0.058 |

---

> ### Author Response · Authors · 2024-11-20
> **Response to Reviewer Rnto**
>
> Table 7: Complexity comparison among different SD-based SR methods. All methods are tested on the ×4 (128→512) SR tasks, and the inference time is measured on an V100 GPU.
> |  Method | StableSR | PASD | SeeSR | SeeSR+UniPC | SeeSR+ DPMsolver | AddSR | OSEDiff | TAD-SR |
> |:-------:|:--------:|:----:|:--------:|:-----:|:-----------:|:----------------:|:-----:|:-----:|
> |   NFE   |    200   |  20  |   50  |      10     |        10        |   1   |   1   |    1    |
> | Inference time (s) |   17.76  |   13.51    |  8.4  |     2.14    |       2.13       |  0.64 |   0.48  |  0.64  |
>
> >**Q8**: Are there any limit conditions for using the method? The author should discuss and analyze the limitations of the proposed method. It is recommended to add a discussion of a discussion of potential limitations or where the proposed method might not perform as well.
>
> >**A8**: Thank you for your suggestion. Although our single-step method demonstrates strong performance, it shares a common limitation with current single-step distillation methods: increasing the number of inference steps alone does not yield better performance. Thus, developing a distillation method that matches the performance of state-of-the-art single-step approaches while enabling additional inference steps to enhance performance is a key area of our ongoing research.
>
> References
>
> [1]Yin, T., Gharbi, M., Zhang, R., Shechtman, E., Durand, F., Freeman, W. T., & Park, T. (2024). One-step diffusion with distribution matching distillation. In Proceedings of the IEEE/CVF Conference on Computer Vision and Pattern Recognition (pp. 6613-6623).
>
> [2]Hertz, A., Aberman, K., & Cohen-Or, D. (2023). Delta denoising score. In Proceedings of the IEEE/CVF International Conference on Computer Vision (pp. 2328-2337).

---

> > ### Comment · Reviewer_B832 · 2024-11-27
> > **Thanks for your response and detailed results**
> >
> > Thank you for your response. I choose to keep my score as is mainly because the performance improvement appears to be somewhat marginal (or, in some cases, the improvement in certain metrics comes at the cost of others), which also validates my previous concerns.

---

> > ### Comment · Reviewer_Rnto · 2024-11-28
> >
> > I appreciate the response from the authors but I will keep my score. The author didn't fully address my concerns.
> > First, the explanation that the proposed model relies heavily on the teacher model does not convince me, and the authors did not explain the efficiency of the proposed method.
> > Second, the author did not provide an intuitive reason for choosing the weighting function and how it affects the training process or results.
> > More importantly, complexity comparison shouldn't just compare the inference time, but should include other key parameters for diffusion models, such as parameters, sampling steps, and MACs.

---

> ### Author Response · Authors · 2024-11-29
> **Response to Reviewer Rnto (Part II)**
>
> Thank you for your response. We will address your remaining concerns as follows.
>
> >Q1: The explanation that the proposed model relies heavily on the teacher model.
>
> >A1: First, we would like to clarify that during inference, only the student model performs single-step sampling to generate samples, while the teacher model supervises the student by generating samples through multi-step sampling during training.
> Second, the knowledge distillation technique aims to transfer knowledge from the teacher model to student model through training, meaning the performance of the student model is inevitably influenced by the teacher. However, to prevent the student model's performance from being entirely constrained by the teacher, we have incorporated ground truth into the distillation framework through adversarial learning, providing additional supervision. Experimental results demonstrate that our method even outperforms the teacher model on certain non-reference metrics. Furthermore, in response to the reviewer’s comments, we replaced the teacher model in our experiments. As shown in Tables 11, 12, and 13 of the paper, our method continues to generate high-quality images through single-step inference, clearly demonstrating its effectiveness.
>
> >Q2: The weighting function of HSD.
>
> >A2: Regarding the loss weight, we followed the approach used in DMD[1] and DDS[2], normalizing the loss across both spatial and channel dimensions($i.e.,\omega = 1/CS$). This normalization is commonly applied in prior model training, as it facilitates better model optimization. We have also provided results without weighting function $\omega$ for comparison, and the effectiveness of the weighting function is evident from Table 1.
>
> >Q3: Complexity comparison.
>
> >A3: Ultimately, we would like to emphasize that both Table 2 in the initial manuscript and Table 6 in the revised manuscript provide a comparison of the sampling steps, inference time, and parameter count between our method and other methods. Additionally, in response to the reviewer’s comments, we have included a comparison of FLOPs, with the results shown in Tables 2 and 3. Table 2 focuses on comparisons with diffusion-based super-resolution methods trained from scratch. Table 3 highlights a comparison of computational complexity with SD-based super-resolution methods.
>
> Table 1: Ablation studies of the weighting function of HSD on RealSR and RealSet65 benchmarks. The best results are highlighted in bold.
> |     Datasets    | RealSet65 | RealSet65 |  RealSR | RealSR |
> |---------------|:---------:|:---------:|:-------:|:------:|
> |     Settings    |  CLIPIQA $\uparrow$ |   MUSIQ  $\uparrow$ | CLIPIQA $\uparrow$ |  MUSIQ $\uparrow$|
> | w/o weighting function |   0.723   |   66.242  |  0.731  | 64.425  |
> |     Ours    |   **0.734**   |    **67.500**   |  **0.741**  | **65.701** |
>
>
> Table 2: Complexity comparison among different SR methods. All methods are tested on the ×4 (64→256) SR tasks, and the inference time is measured on an A100 GPU.
> |  Method  |  LDM  | ResShift (teacher) | SinSR | DMD |  TAD-SR |
> |-------|:------:|:-----------:|:------:|:------:|:----------:|
> |   NFE   |    15   |      15      |    1   |    1   |      1     |
> |   #Parameters (M)   |    168.92   |      173.91      |    173.91    |   173.91    |      173.91      |
> | Inference time (s) |  0.408 |    0.682    |  0.058  | 0.058 |  0.058 |
> |FLOPs (G)|  1208.7 | 1506.75 | 100.45 | 100.45 | 100.45 |
>
> Table 3: Complexity comparison among different SD-based SR methods. All methods are tested on the ×4 (128→512) SR tasks, and the inference time is measured on an V100 GPU.
> |  Method | StableSR | PASD | SeeSR (teacher) | AddSR | OSEDiff | TAD-SR |
> |-------|:--------:|:----:|:--------:|:----------------:|:-----:|:-----:|
> |   NFE   |    200   |  20  |   50  |      1     |        1        |   1   |
> |   #Parameters (M)   |    1002.95  |  1333.53      |        1703.05        |   1703.05  |   1378.39  |    1703.05    |
> | Inference time (s)|   17.76  |   13.51    |  8.4     |  0.64 |   0.48  |  0.64  |
> | FLOPs (G) |   157294  |   28675.2    |  71148     |  8488.76 |   7995.5  |  8488.76 |
>
> [1]Yin, T., Gharbi, M., Zhang, R., Shechtman, E., Durand, F., Freeman, W. T., & Park, T. (2024). One-step diffusion with distribution matching distillation. In Proceedings of the IEEE/CVF Conference on Computer Vision and Pattern Recognition (pp. 6613-6623).
>
> [2]Hertz, A., Aberman, K., & Cohen-Or, D. (2023). Delta denoising score. In Proceedings of the IEEE/CVF International Conference on Computer Vision (pp. 2328-2337).

---

> > ### Comment · Reviewer_Rnto · 2024-12-03
> >
> > Thank you for your response. I have no further questions and am willing to increase my score.

---

> > > ### Author Response · Authors · 2024-12-03
> > > **Thank you for your response**
> > >
> > > Thank you for carefully reviewing the discussion and deciding to increase your score. We are pleased to revise the manuscript based on your suggestions, which have made it more robust and easier to understand.

---

### Official Review · Reviewer_B832 · 2024-11-03

**Soundness:** 2
**Presentation:** 2
**Contribution:** 2
**Rating:** 5
**Confidence:** 4

**Summary:**

This paper introduces TAD-SR, a time-aware diffusion distillation method designed to enhance the efficiency and performance of diffusion-based image super-resolution (SR) models. By aligning the student and teacher models with the proposed score distillation strategy and incorporating a time-aware discriminator to distinguish real and synthetic data across varying noise levels, TAD-SR achieves strong performance across several metrics.

**Strengths:**

1. The topic is interesting and meaningful.
2. Extensive experiments demonstrate that TAD-SR achieves results comparable to or exceeding multi-step diffusion models, espeically in some non-reference IQA metrics.

**Weaknesses:**

1. The organization of the paper needs improvement, as it is challenging to clearly understand the core idea. For instance, Fig. 2, which aims to illustrate the paper's motivation, has a caption that provides limited information.

2. The paper lacks essential metrics, such as PSNR and SSIM, to evaluate model fidelity. As shown in previous works, there is a trade-off between PSNR, SSIM, and CLIPIQA, MUSIQ. Reporting only LPIPS and non-reference IQA metrics is insufficient to demonstrate performance. Both the main results and ablation studies should include these metrics.

3. Although I understand that StableDiffusionXL also employs adversarial loss, it appears less elegant to me due to the inherent limitations of GANs.

4. In addition to the difficulty of assessing performance without PSNR and SSIM, the reported improvements seem marginal compared to existing methods.

**Questions:**

The motivation is not clear. If the proposed method wants to achieve one-step SR, why it is important for student model to learn how to deal with the intermediate steps?

Will increase the inference steps contribute to the improvement of the performance?

---

> ### Author Response · Authors · 2024-11-20
> **Response to Reviewer B832**
>
> Thank you for your comments and feedback. We address your concerns here.
>
> >**Q1**: The organization of the paper needs improvement, as it is challenging to clearly understand the core idea. For instance, Fig. 2, which aims to illustrate the paper's motivation, has a caption that provides limited information.
>
> >**A1**: Thank you for your suggestion. We will carefully describe the details of this method in the revised manuscript to improve the readability and clarity of the paper.
>
> >**Q2**: The paper lacks essential metrics, such as PSNR and SSIM, to evaluate model fidelity. As shown in previous works, there is a trade-off between PSNR, SSIM, and CLIPIQA, MUSIQ. Reporting only LPIPS and non-reference IQA metrics is insufficient to demonstrate performance. Both the main results and ablation studies should include these metrics.
>
> >**A2**: Thank you for your suggestion. We have included PSNR and SSIM metrics in both our main experiments and ablation studies, as shown in Tables 1, 2, and 3. However, our experimental results, along with findings from previous studies, indicate that PSNR and SSIM do not always align with human perception or other indicators such as LPIPS, CLIPIQA, and MUSIQ. Specifically, when image quality improves, and these perceptual indicators yield higher values, PSNR and SSIM often decrease. Conversely, an increase in PSNR and SSIM typically corresponds to smoother and blurrier images. For instance, while methods such as LDM, ResShift, and DASR achieve higher PSNR and SSIM scores compared to others, the images they generate tend to appear smoother or blurrier (as shown in Figures 6 and 12). We infer that this discrepancy likely arises because PSNR and SSIM measure image differences in pixel space, whereas human perception and other metrics evaluate images based on perceptual quality. Therefore, we regard PSNR and SSIM as reference metrics rather than primary evaluation metrics in real-world super-resolution tasks, consistent with the conclusions of prior work [1][2][3].
>
> Table 1: Quantitative results of different methods on the dataset of ImageNet-Test. The best and second best results are highlighted in bold and italic. ∗ indicates that the result was obtained by replicating the method in the paper.
> |   Methods   |    PSNR $\uparrow$   |    SSIM  $\uparrow$ |   LPIPS  $\downarrow$ |  CLIPIQA $\uparrow$ |    MUSIQ $\uparrow$  |
> |-----------|:---------:|:---------:|:---------:|:---------:|:----------:|
> |    ESRGAN   |   20.67   |   0.448   |   0.485   |   0.451   |   43.615   |
> | RealSR-JPEG |   23.11   |   0.591   |   0.326   |   0.537   |   46.981   |
> |    BSRGAN   |   24.42   |   0.659   |   0.259   |   0.581   |  _54.697_  |
> |    SwinIR   |   23.99   |   0.667   |   0.238   |   0.564   |    53.790   |
> |  RealESRGAN |   24.04   |   0.665   |   0.254   |   0.523   |   52.538   |
> |     DASR    |   24.75   |  _0.675_  |    0.250   |   0.536   |   48.337   |
> |    LDM-15   |  _24.89_  |    0.670   |   0.269   |   0.512   |   46.419   |
> | ResShift-15 | **25.01** | **0.677** |   0.231   |   0.592   |    53.660   |
> |   SinSR-1   |   24.56   |   0.657   | **0.221** |   0.611   |   53.357   |
> |   SinSR*-1  |   24.59   |   0.659   |   0.231   |   0.599   |   52.462   |
> |    DMD*-1   |   24.05   |   0.629   |   0.246   |  _0.612_  |   54.124   |
> |   TAD-SR-1  |   23.91   |   0.641   |  _0.227_  | **0.652** | **57.533** |

---

> ### Author Response · Authors · 2024-11-20
> **Response to Reviewer B832**
>
> Table 2: Quantitative results of different methods on the dataset of CelebA-Test. The best and second best results are highlighted in bold and italic. ∗ indicates that the result was obtained by replicating the method in the paper.
> |    Methods    |    PSNR $\uparrow$   |   SSIM $\uparrow$   |   LPIPS $\downarrow$  |    IDS $\downarrow$    |    LMD $\downarrow$   |   FID-F $\downarrow$   |   FID-G $\downarrow$   |  CLIPIQA $\uparrow$ |  MUSIQ $\uparrow$  |
> |-------------|:----------:|:---------:|:---------:|:----------:|:---------:|:----------:|:----------:|:---------:|:--------:|
> |     DFDNET    |   10.833   |   0.449   |   0.739   |   86.323   |   20.784  |   93.621   |   76.118   |   0.619   |  51.173  |
> |    PSFRGAN    |   19.662   |   0.582   |   0.475   |   74.025   |   10.168  |   63.676   |   60.748   |    0.630   |   69.910  |
> |   GFPGANv1.2  |   19.558   |   0.605   |   0.416   |    66.820   |   8.886   |   66.308   |   27.698   |   0.671   | _75.388_ |
> | RestoreFormer |   19.604   |   0.551   |   0.488   |   70.518   |   11.137  |   50.165   |   51.997   | **0.736** |  71.039  |
> |      VQFR     |   19.979   |   0.622   |   0.411   |   65.538   |    8.910   |   58.423   |   25.234   |   0.685   |  73.155  |
> |  CoderFormer  |  _23.576_  |   0.661   |   0.324   | **59.136** |   5.035   |   62.794   |    26.160   |   0.698   | **75.900** |
> |  DiffFace-100 | **24.033** | **0.705** |   0.338   |   63.033   |   5.301   |   52.531   |   23.212   |   0.527   |  66.042  |
> |  Resshift-15  |   23.413   |  _0.671_  | **0.309** |  _59.623_  |   5.056   |   50.164   |   17.564   |   0.613   |  73.214  |
> |    SinSR*-1   |   22.317   |    0.640   |   0.319   |   60.305   | **4.935** |   55.292   |   21.681   |   0.634   |   74.140  |
> |    TAD-SR-1   |   22.614   |   0.629   |   0.341   |   59.897   |   _5.050_  | **41.968** | **16.779** |  _0.735_  |  75.027  |
>
> Table 3: Ablation studies of the proposed methods on ImageNet-Test benchmarks. The best results are highlighted in bold.
>
> | Score   distillation | Discriminator |    PSNR $\uparrow$  |    SSIM $\uparrow$  |   LPIPS $\downarrow$  |  CLIPIQA $\uparrow$ |    MUSIQ $\uparrow$  |
> |:--------------------:|:-------------:|:---------:|:---------:|:---------:|:---------:|:----------:|
> |          SDS         |       ✘    |   24.46  |   0.658   |   0.335   |   0.412   |    41.133   |
> |          SDS         |       ✔       | **24.76** |    0.670   |    0.300   |   0.469   |    46.024  |
> |          SDS         |   time-aware  |   24.69   | **0.671** |   0.278   |   0.522   |    49.932   |
> |          HSD         |       ✘       |   24.64  |   0.661   |   0.228   |   0.608   |   53.508   |
> |          HSD         |       ✔       |   23.89  |    0.640   | **0.227** |   0.649   |    57.370   |
> |          HSD         |   time-aware  |   23.91  |   0.641   | **0.227** | **0.652** | **57.533** |
>
> >**Q3**: Although I understand that StableDiffusionXL also employs adversarial loss, it appears less elegant to me due to the inherent limitations of GANs.
>
> >**A3**: Recently, many diffusion-based methods [4][5] have begun integrating adversarial learning into the training process. Experimental results demonstrate that this approach can significantly enhance model performance, underscoring its potential value.
>
> >**Q4**: In addition to the difficulty of assessing performance without PSNR and SSIM, the reported improvements seem marginal compared to existing methods.
>
> >**A4**: In addition to PSNR and SSIM, our method demonstrates significant improvements over SinSR in other metrics. The table below lists the percentage improvements achieved by our method compared to SinSR.
>
> Table4: Quantitative comparison with SinSR method in super-resolution tasks.
> | Datasets |              | ImageNet-Test |               |   RealSR    |       RealSR         |  RealSet65 |     RealSet65        |
> |--------|:------------:|:-------------:|:-------------:|:------------:|:-------------:|:----------:|:-----------:|
> |  Method  |     LPIPS  $\downarrow$  |    CLIPIQA  $\uparrow$  |     MUSIQ   $\uparrow$  |    CLIPIQA $\uparrow$  |     MUSIQ   $\uparrow$  |   CLIPIQA $\uparrow$ |    MUSIQ  $\uparrow$  |
> |  SinSR*  |     0.231    |     0.599     |     52.462    |     0.691    |     60.865    |    0.712   |    62.575   |
> |  TAD-SR  | 0.227(+1.7%) |  0.652(+8.8%) | 57.533(+9.7%) | 0.741(+7.2%) | 65.701(+7.9%) | 0.734(+3%) | 67.5(+7.9%) |

---

> ### Author Response · Authors · 2024-11-20
> **Response to Reviewer B832**
>
> >**Q5**: The motivation is not clear. If the proposed method wants to achieve one-step SR, why it is important for student model to learn how to deal with the intermediate steps?
>
> >**A5**: Sorry, it may be that our description was not clear enough, which caused a misunderstanding for you. We will enhance the readability of the paper in the revised PDF. To clarify, our student model accepts a fixed time step $T$ to generate clean samples in a single step. The intermediate time steps we sample are used solely to calculate the loss. Specifically, we leverage the pre-trained diffusion model's ability to handle intermediate time steps to constrain the single-step output of the student model. Diffusion models typically predict low-frequency information in the early stages of denoising and high-frequency information in the later stages. Therefore, we add varying levels of noise to both the clean samples generated by the student model and the teacher model, then feed them into a pre-trained diffusion model for prediction. By calculating the distance between the two predicted values, we can constrain the samples generated by the student model to match the high-frequency or low-frequency information in the teacher model's generated samples.
>
> >**Q6**： Will increase the inference steps contribute to the improvement of the performance?
>
> >**A6**: This is really a good question! Normally, if only a single time step is sampled to train the student model, simply increasing the number of iterations during inference will not lead to any performance improvement. This is because the model has only learned the mapping from noisy data to clean data at that specific time step and lacks the ability to process noisy data at other intermediate time steps. This limitation is common to all single-step distillation methods. Thus, developing a distillation method that matches the performance of state-of-the-art single-step approaches while enabling additional inference steps to enhance performance is a key area of our ongoing research. We will include a discussion on this aspect in the revised PDF.
>
> References
>
> [1] Wang, J., Yue, Z., Zhou, S., Chan, K. C., & Loy, C. C. (2024). Exploiting diffusion prior for real-world image super-resolution. International Journal of Computer Vision, 1-21.
>
> [2] Xie, R., Tai, Y., Zhao, C., Zhang, K., Zhang, Z., Zhou, J., ... & Yang, J. (2024). Addsr: Accelerating diffusion-based blind super-resolution with adversarial diffusion distillation. arXiv preprint arXiv:2404.01717.
>
> [3] Wu, R., Yang, T., Sun, L., Zhang, Z., Li, S., & Zhang, L. (2024). Seesr: Towards semantics-aware real-world image super-resolution. In Proceedings of the IEEE/CVF conference on computer vision and pattern recognition (pp. 25456-25467).
>
> [4] Sauer, A., Lorenz, D., Blattmann, A., & Rombach, R. (2025). Adversarial diffusion distillation. In European Conference on Computer Vision (pp. 87-103). Springer, Cham.
>
> [5] Xu, Y., Zhao, Y., Xiao, Z., & Hou, T. (2024). Ufogen: You forward once large scale text-to-image generation via diffusion gans. In Proceedings of the IEEE/CVF Conference on Computer Vision and Pattern Recognition (pp. 8196-8206).

---

> > ### Author Response · Authors · 2024-12-04
> > **Response to Reviewer B832**
> >
> > Dear reviewer B832
> >
> > We sincerely appreciate your response, but it seems that you have replied in the wrong place. We will continue to address your concerns here.
> >
> > Our method demonstrates significant improvements over other methods in most metrics for real-world image super-resolution and blind face restoration tasks, particularly when compared to SinSR, a single-step SR technique. Additionally, we replaced the teacher model (ResShift) with an SD-based SR model (SeeSR) and conducted extensive experiments. The experimental results are presented in Tables 11, 12, and 13 of the manuscript. Our method achieved performance comparable to the teacher model and outperformed other comparison methods in most metrics, effectively validating the effectiveness of our approach. Furthermore, it is noteworthy that previous methods were also unable to consistently outperform comparative methods across all indicators and scenarios, which is a highly challenging task.

---

### Official Review · Reviewer_uBAa · 2024-11-05

**Soundness:** 3
**Presentation:** 3
**Contribution:** 3
**Rating:** 5
**Confidence:** 5

**Summary:**

This  paper proposes a time-aware diffusion distillation method, TAD-SR, to achieve one-step SR inference with competitive performance. It applies a score distillation strategy make efforts to eliminate the inherent bias SDS focus more on high-frequency image details when sampling at small time steps. A time-aware discriminator is also designed to differentiate between real and synthetic data.

**Strengths:**

1.	This paper proposes a time-aware distillation method that accelerates diffusion-based SR models into a single inference step.
2.	The writing of this paper is good.

**Weaknesses:**

See the questions.

**Questions:**

1.	Since this is a distillation method, please compare more diffusion-based distillation SR methods, like OSEDiff [1], quantitatively and qualitatively. (Why are the comparison with diffusion-based distillation SR methods missing in some tables and figures?)

2.	Since you claim that TAD-SR can achieve better reconstruction of high-frequency information, please present the spectrum images of the LR input, GT, baseline methods’ reconstruction, and TAD-SR’s reconstruction. Examine the differences in the high-frequency patterns around the periphery of the spectrum images.

3.	Please compare the inference time of TAD-SR and baseline methods.

4.	In Fig. 10 and Fig. 12, TAD-SR’s results appear to contain many fragmented particles, which make the images look sharper at first glance; however, this is actually due to the addition of pseudo-textures or unnatural details. Could you explain the cause of this? For instance, could it be due to the adversarial loss?

5.	Following the concern raised in my 4th question, could you please provide more qualitative  comparisons that contain fine details or small textures?

[1] Rongyuan Wu, et al. One-Step Effective Diffusion Network for Real-World Image Super-Resolution.


(I apologize for my previous review comments, which were not fully aligned with your article due to a heavy review workload. I am providing corrected feedback here, and if your response addresses these points well, I will consider adjusting the score.)

---

> ### Author Response · Authors · 2024-11-20
> **Response to Reviewer uBAa**
>
> Thank you for providing valuable feedback on our paper despite your busy schedule. We address your concerns here.
>
> >**Q1**: Since this is a distillation method, please compare more diffusion-based distillation SR methods, like OSEDiff [1], quantitatively and qualitatively. (Why are the comparison with diffusion-based distillation SR methods missing in some tables and figures?)
>
> >**A1**: Thank you for pointing out this issue. We have compared our method with OSEDiff, with quantitative results presented in Tables 1, 2, and 3. In the main text, we primarily use the super-resolution model ResShift trained from scratch as the teacher, enabling a fair comparison with SinSR, which also distills ResShift. In the appendix, we employ the SD-based SR method SeeSR as the teacher and mainly compare our approach with other SD-based SR methods and SD-based distillation SR methods. Due to substantial differences in the datasets used to train ResShift and SeeSR, comparisons involving SD-based SR methods are omitted from certain charts in the main text.
>
> Table 1: Quantitative comparison with state of the arts on RealSR dataset. Following the experimental setup of SeeSR, the LR images in the RealSR dataset were center-cropped to 128 $\times$ 128. The best and second best results are highlighted in bold and italic.
> |        Methods       |    PSNR  $\uparrow$  |   LPIPS $\downarrow$  |     FID  $\downarrow$  |   NIQE  $\downarrow$ |  CLIPIQA $\uparrow$  |   MUSIQ  $\uparrow$  |  MANIQA $\uparrow$  |
> |--------------------|:---------:|:---------:|:----------:|:--------:|:---------:|:---------:|:--------:|
> |   BSRGAN   | _26.49_ | **0.267** | 141.28 | 5.66 |  0.512  | 63.28 |  0.376 |
> | RealESRGAN | 25.78 | _0.273_ | 135.18 | 5.83 |  0.449  | 60.36 |  0.373 |
> |     LDL    | 25.09 | 0.277 | 142.71 |   6.00  |   0.430  | 58.04 |  0.342 |
> |   FeMaSR   | 25.17 | 0.294 | 141.05 | 5.79 |  0.541  | 59.06 |  0.361 |
> |     StableSR-200     |   25.63   |   0.302   |    133.40   |   5.76  |   0.528   |   61.11   |   0.366  |
> |      ResShift-15     |   26.34   |   0.346   |   149.54   |   6.87  |   0.542   |   56.06   |   0.375  |
> |        PASD-20       | **26.67** |   0.344   |   _122.30_  |   6.06  |   0.519   |   62.92   |   0.404  |
> |       SeeSR-50       |   25.24   |   0.301   |   125.42   |  _5.39_  |   _0.670_  | **69.82** | **0.540** |
> |    SeeSR(UniPC-10)   |   25.86   |   0.281   |   122.41   |   5.53  |   0.577   |   67.12   |   0.476  |
> | SeeSR(DPMSolver-10) |    25.90   |   0.281   |   122.46   |   5.54  |   0.581   |   67.12   |   0.478  |
> |        SinSR-1       |   26.16   |   0.308   |   142.44   |   5.75   |    0.630   |   60.96   |   0.399  |
> |        AddSR-1       |   23.12   |   0.309   |   132.01   |   5.54   |   0.552   |   67.14   |   0.488  |
> |       OSEDiff-1      |   25.15   |   0.292   |   123.49   |   5.63   |   0.668   |   68.99   |   0.474  |
> |       TAD-SR-1       |    24.50   |   0.304   | **118.38** | **5.13** | **0.676** |  _69.02_  |  _0.526_ |
>
> Table 2: Quantitative comparison with state of the arts on RealLR200 dataset dataset. The best and second best results are highlighted in bold and italic. Note that since the RealLR200 dataset lacks high-resolution images, we only computed non-reference metrics.
>
> |        Methods       |   NIQE $\downarrow$  |  CLIPIQA $\uparrow$ |   MUSIQ  $\uparrow$ |   MANIQA $\uparrow$ |
> |--------------------|:--------:|:---------:|:---------:|:---------:|
> |   BSRGAN   | 4.38 |   0.570  | 64.87 |  0.369 |
> | RealESRGAN |  4.20 |  0.542  | 62.93 |  0.366 |
> |     LDL    | 4.38 |  0.509  | 60.95 |  0.327 |
> |   FeMaSR   | 4.34 |  0.655  | 64.24 |  0.410  |
> |     StableSR-200     |   4.25   |   0.592   |   62.89   |   0.367   |
> |      ResShift-15     |   6.29   |   0.647   |   60.25   |   0.418   |
> |        PASD-20       |   4.18   |   0.620   |   66.35   |   0.419   |
> |       SeeSR-50       |   4.16   |   0.662   |   68.63   | **0.491** |
> |    SeeSR(UniPC-10)   |   4.25   |   0.601   |    66.90   |   0.433   |
> | SeeSR(DPMSolver-10) |   4.28   |   0.603   |   66.92   |   0.435   |
> |        SinSR-1       |   5.62   | **0.697** |   63.85   |   0.445   |
> |        AddSR-1       |   4.06   |   0.585   |   66.86   |   0.418   |
> |       OSEDiff-1      |  _4.05_  |  _0.674_  | **69.61** |   0.444   |
> |       TAD-SR-1       | **3.95** |  _0.674_  |  _69.48_  |  _0.482_  |

---

> ### Author Response · Authors · 2024-11-20
> **Response to Reviewer uBAa**
>
> Table 3: Quantitative comparison with state of the arts on DIV2k-val dataset. The best and second best results are highlighted in bold and italic.
>
> |        Methods       |    PSNR $\uparrow$ |   LPIPS $\downarrow$  |    FID $\downarrow$   |   NIQE  $\downarrow$ |  CLIPIQA $\uparrow$ |   MUSIQ  $\uparrow$ |   MANIQA $\uparrow$ |
> |--------------------|:---------:|:---------:|:---------:|:--------:|:---------:|:---------:|:---------:|
> |   BSRGAN   | _24.58_ | 0.335 | 44.22 | 4.75 |  0.524  | 61.19 |  0.356 |
> | RealESRGAN | 24.29 | _0.311_ | 37.64 | 4.68 |  0.527  | 61.06 |  0.382 |
> |     LDL    | 23.83 | 0.326 | 42.28 | 4.86 |  0.518  | 60.04 |  0.375 |
> |   FeMaSR   | 23.06 | 0.346 |  53.7 | 4.74 |  0.599  | 60.82 |  0.346 |
> |     StableSR-200     |   23.29   |   0.312   | **24.54** |   4.75   |  _0.676_  |   65.83   |   0.422   |
> |      ResShift-15     | **24.72** |    0.34   |   41.99   |   6.47   |   0.594   |   60.89   |   0.399   |
> |        PASD-20       |   24.51   |   0.392   |   31.58   |   5.37   |   0.551   |   59.99   |   0.399   |
> |       SeeSR-50       |   23.68   |   0.319   |   25.97   |   4.81   | **0.693** | **68.68** | **0.504** |
> |    SeeSR(UniPC-10)   |   24.07   |   0.339   |   27.33   |   5.00   |   0.607   |   64.97   |   0.432   |
> | SeeSR(DPMSolver-10) |   24.12   |   0.338   |   27.32   |   5.03   |   0.612   |   65.07   |   0.435   |
> |        SinSR-1       |   24.41   |   0.324   |   35.23   |   6.01   |   0.648   |   62.80   |   0.424   |
> |        AddSR-1       |   23.26   |   0.362   |   29.68   |   4.76   |   0.573   |   63.69   |   0.405   |
> |       OSEDiff-1      |   23.72   | **0.294** |   26.33   |  _4.71_  |   0.661   |  _67.96_  |   0.443   |
> |       TAD-SR-1       |   23.54   |  _0.311_  |  _25.96_  | **4.64** |   0.664   |   67.01   |  _0.470_  |
>
> >**Q2**: Since you claim that TAD-SR can achieve better reconstruction of high-frequency information, please present the spectrum images of the LR input, GT, baseline methods’ reconstruction, and TAD-SR’s reconstruction. Examine the differences in the high-frequency patterns around the periphery of the spectrum images.
>
> >**A2**: Thank you for your valuable suggestion. In Figure 10 of the appendix, we present the Fourier transform spectra of low-resolution (LR) images, ground truth (GT) images, and reconstructions from different super-resolution (SR) methods. From these spectra, it is evident that our method preserves more high-frequency information compared to other diffusion-based SR methods.

---

> ### Author Response · Authors · 2024-11-20
> **Response to Reviewer uBAa**
>
> >**Q3**: Please compare the inference time of TAD-SR and baseline methods.
>
> >**A3**: Based on the reviewers' feedback, we have included a complexity comparison between TAD-SR and baseline methods, as presented in Tables 4 and 5. Table 4 focuses on comparisons with GAN-based methods and diffusion-based super-resolution methods trained from scratch. The results demonstrate that TAD-SR accelerates the teacher model, ResShift, to a single inference step, improving its speed by approximately tenfold. Table 5 highlights a comparison of inference time with SD-based super-resolution methods, revealing that our method's inference delay is only 7.6% of the teacher model, SeeSR.
>
> Table 4: Complexity comparison among different SR methods. All methods are tested on the ×4 (64→256) SR tasks, and the inference time is measured on an A100 GPU.
> |  Method | ESRGAN | RealSR-JPEG | BSRGAN | SwinIR | RealESRGAN |  DASR |  LDM  | ResShift | SinSR | TAD-SR |
> |-------|:------:|:-----------:|:------:|:------:|:----------:|:-----:|:-----:|:--------:|:-----:|:------:|
> |   NFE   |    1   |      1      |    1   |    1   |      1     |   1   |   15  |    15    |   1   |    1   |
> | Inference time (s) |  0.038 |    0.038    |  0.038 |  0.107 |    0.038   | 0.022 | 0.408 |   0.682  | 0.058 |  0.058 |
>
> Table 5: Complexity comparison among different SD-based SR methods. All methods are tested on the ×4 (128→512) SR tasks, and the inference time is measured on an V100 GPU.
> |  Method | StableSR | PASD | SeeSR | SeeSR+UniPC | SeeSR+ DPMsolver | AddSR | OSEDiff | TAD-SR |
> |-------|:--------:|:----:|:--------:|:-----:|:-----------:|:----------------:|:-----:|:-----:|
> |   NFE   |    200   |  20  |   50  |      10     |        10        |   1   |   1   |    1    |
> | Inference time (s) |   17.76  |   13.51    |  8.4  |     2.14    |       2.13       |  0.64 |   0.48  |  0.64  |
>
> >**Q4**: In Fig. 10 and Fig. 12, TAD-SR’s results appear to contain many fragmented particles, which make the images look sharper at first glance; however, this is actually due to the addition of pseudo-textures or unnatural details. Could you explain the cause of this? For instance, could it be due to the adversarial loss?
>
> >**A4**: Upon careful examination of the images generated by our method and other super-resolution approaches, we observed that various methods may produce pseudo-textures in certain images to differing extents. We found that some of the unnatural textures generated by our method exhibit the same pattern as those produced by the teacher model, which we speculate may be due to the inherent properties of the diffusion model itself. Additionally, on real-world datasets, this phenomenon is likely attributed to the inconsistent degradation encountered by the training and testing models. While degradation during model training is artificially synthesized and may exhibit certain statistical features, real-world degradation is more complex and diverse, which could lead to the generation of pseudo-textures.
>
> >**Q5**: Following the concern raised in my 4th question, could you please provide more qualitative comparisons that contain fine details or small textures?
>
> >**A5**: Sure, we provide more qualitative comparisons that contain fine details in Figure 9 and Figure 15 of the revised PDF.

---

> > ### Author Response · Authors · 2024-12-03
> > **Official Comment by Authors**
> >
> > Dear Reviewer uBAa:
> >
> > The discussion period between the authors and the reviewer is nearing its end, and we kindly request that you review our clarifications and revisions. If our response addresses your concerns, we hope you can reconsider your score.
> >
> > Thank you once again for your time and consideration.
> >
> > Best Wishes!
> >
> > Authors of Submission 1713

---

### Official Review · Reviewer_gXos · 2024-11-05

**Soundness:** 3
**Presentation:** 3
**Contribution:** 3
**Rating:** 6
**Confidence:** 3

**Summary:**

This paper introduces a time-aware diffusion distillation method named TAD-SR, which enables the student model to focus on high-frequency image details at smaller time steps and eliminates inherent biases in score distillation sampling. The authors also design a time-aware discriminator that fully leverages the teacher model’s knowledge by injecting time information to differentiate between real and synthetic data. Experimental results demonstrate the effectiveness and efficiency of the proposed method.

**Strengths:**

* The paper is well-written.
* Experimental results demonstrate that the proposed method achieves state-of-the-art performance with high efficiency.

**Weaknesses:**

* The evaluation is not comprehensive. Some image fidelity metrics are lacking, such as PSNR and SSIM on ImageNet-Test, where the competing methods ResShift and SinSR all reported.

* The improvement over the previous single-step distillation method SinSR is minor. Considering that LPIPS—a crucial metric for perceptual quality—is very important, the increase from 0.221 to 0.227 represents a big drop in quality and is not slight.

* The ablation study examines only the presence or absence of the discriminator, neglecting other important aspects—for example, the number of scales used in the discriminator.

**Questions:**

Please refer to the weakness part.

---

> ### Author Response · Authors · 2024-11-20
> **Response to  Reviewer gXos**
>
> Thank you for your comments and feedback. We address your concerns here.
>
> >**Q1**: The evaluation is not comprehensive. Some image fidelity metrics are lacking, such as PSNR and SSIM on ImageNet-Test, where the competing methods ResShift and SinSR all reported.
>
> >**A1**: Thank you for your suggestion. We incorporate PSNR and SSIM metrics into the evaluation of the ImageNet dataset. However, we want to emphasize that these two metrics are secondary in real-world super-resolution tasks[1][2][3]. For instance, while methods such as LDM, ResShift, and DASR achieve higher PSNR and SSIM scores compared to others, the images they generate tend to appear smoother or blurrier (as shown in Figures 6 and 12). This discrepancy likely arises because PSNR and SSIM measure image differences in pixel space, whereas human and other metrics evaluate images based on perceptual quality. Therefore, PSNR and SSIM metrics should be considered as reference points only, which aligns with observations in previous studies[1][2][3].
>
> Table 1: Quantitative results of different methods on the dataset of ImageNet-Test. The best and second best results are highlighted in bold and italic. ∗ indicates that the result was obtained by replicating the method in the paper.
> |   Methods   |    PSNR $\uparrow$   |    SSIM  $\uparrow$ |   LPIPS  $\downarrow$ |  CLIPIQA $\uparrow$ |    MUSIQ $\uparrow$  |
> |-----------|:---------:|:---------:|:---------:|:---------:|:----------:|
> |    ESRGAN   |   20.67   |   0.448   |   0.485   |   0.451   |   43.615   |
> | RealSR-JPEG |   23.11   |   0.591   |   0.326   |   0.537   |   46.981   |
> |    BSRGAN   |   24.42   |   0.659   |   0.259   |   0.581   |  _54.697_  |
> |    SwinIR   |   23.99   |   0.667   |   0.238   |   0.564   |    53.790   |
> |  RealESRGAN |   24.04   |   0.665   |   0.254   |   0.523   |   52.538   |
> |     DASR    |   24.75   |  _0.675_  |    0.250   |   0.536   |   48.337   |
> |    LDM-15   |  _24.89_  |    0.670   |   0.269   |   0.512   |   46.419   |
> | ResShift-15 | **25.01** | **0.677** |   0.231   |   0.592   |    53.660   |
> |   SinSR-1   |   24.56   |   0.657   | **0.221** |   0.611   |   53.357   |
> |   SinSR*-1  |   24.59   |   0.659   |   0.231   |   0.599   |   52.462   |
> |    DMD*-1   |   24.05   |   0.629   |   0.246   |  _0.612_  |   54.124   |
> |   TAD-SR-1  |   23.91   |   0.641   |  _0.227_  | **0.652** | **57.533** |
> >**Q2**: The improvement over the previous single-step distillation method SinSR is minor. Considering that LPIPS—a crucial metric for perceptual quality—is very important, the increase from 0.221 to 0.227 represents a big drop in quality and is not slight.
>
> >**A2**: Thank you for your feedback. We replicate SinSR using the open-source code, and the experimental evaluation results are shown in the third-to-last row of Table 1. Compared to the replicated SinSR, our method improved the LPIPS metric, decreasing it from 0.231 to 0.227. Additionally, our method demonstrates significant improvements over SinSR in most metrics. The table below lists the percentage improvements achieved by our method compared to SinSR.
>
> Table2: Quantitative comparison with SinSR method in super-resolution tasks.
> | Datasets |              | ImageNet-Test |               |  RealSR    |      RealSR         |  RealSet65 |     RealSet65        |
> |--------|:------------:|:-------------:|:-------------:|:------------:|:-------------:|:----------:|:-----------:|
> |  Method  |     LPIPS  $\downarrow$  |    CLIPIQA  $\uparrow$  |     MUSIQ   $\uparrow$  |    CLIPIQA $\uparrow$  |     MUSIQ   $\uparrow$  |   CLIPIQA $\uparrow$ |    MUSIQ  $\uparrow$  |
> |  SinSR*  |     0.231    |     0.599     |     52.462    |     0.691    |     60.865    |    0.712   |    62.575   |
> |  TAD-SR  | 0.227(+1.7%) |  0.652(+8.8%) | 57.533(+9.7%) | 0.741(+7.2%) | 65.701(+7.9%) | 0.734(+3%) | 67.5(+7.9%) |

---

> > ### Author Response · Authors · 2024-11-20
> > **Response to Reviewer gXos**
> >
> > >**Q3**: The ablation study examines only the presence or absence of the discriminator, neglecting other important aspects—for example, the number of scales used in the discriminator.
> >
> > >**A3**: Thank you for your valuable suggestion. We also conducted ablation experiments to evaluate the impact of using multi-scale features in the discriminator. We designed an experiment using only the features of the last layer of the diffusion model for discrimination, denoted as "w/o multi-scale". Now, our analysis of the discriminator includes comparisons with and without the discriminator, the incorporation of temporal information, and the use of multi-scale features. From Table 3, it can be seen that the discriminator utilizing multi-scale features and incorporating temporal information achieves the best performance.
> >
> > Table 3: Ablation studies of our proposed discriminator on RealSR and RealSet65 benchmarks. The best results are highlighted in bold.
> > |     Datasets    | RealSet65 | RealSet65 |  RealSR | RealSR |
> > |:---------------:|:---------:|:---------:|:-------:|:------:|
> > |     Settings    |  CLIPIQA $\uparrow$ |   MUSIQ  $\uparrow$ | CLIPIQA $\uparrow$ |  MUSIQ $\uparrow$|
> > |     Ours discriminator    |   **0.734**   |    **67.500**   |  **0.741**  | **65.701** |
> > |  w/o time-aware |   0.729   |   66.904  |  0.711  |  63.550 |
> > | w/o multi-scale |   0.724   |   67.330   |  0.722  | 65.205 |
> >
> > References
> >
> > [1] Wang, J., Yue, Z., Zhou, S., Chan, K. C., & Loy, C. C. (2024). Exploiting diffusion prior for real-world image super-resolution. International Journal of Computer Vision, 1-21.
> >
> > [2] Xie, R., Tai, Y., Zhao, C., Zhang, K., Zhang, Z., Zhou, J., ... & Yang, J. (2024). Addsr: Accelerating diffusion-based blind super-resolution with adversarial diffusion distillation. arXiv preprint arXiv:2404.01717.
> >
> > [3] Wu, R., Yang, T., Sun, L., Zhang, Z., Li, S., & Zhang, L. (2024). Seesr: Towards semantics-aware real-world image super-resolution. In Proceedings of the IEEE/CVF conference on computer vision and pattern recognition (pp. 25456-25467).

---

> > > ### Author Response · Authors · 2024-12-03
> > > **Official Comment by Authors**
> > >
> > > Dear Reviewer gXos:
> > >
> > > The discussion period between the authors and the reviewer is nearing its end, and we kindly request that you review our clarifications and revisions. If our response addresses your concerns, we hope you can reconsider your score.
> > >
> > > Thank you once again for your time and consideration.
> > >
> > > Best Wishes!
> > >
> > > Authors of Submission 1713

---

### Official Review · Reviewer_eiDx · 2024-11-07

**Soundness:** 3
**Presentation:** 2
**Contribution:** 3
**Rating:** 5
**Confidence:** 4

**Summary:**

This paper proposed a method to distill a super-resolution diffusion model into one step, by combining 3 losses: direct regression loss, GAN loss, and a modified score distillation loss. The main contribution is the score distillation part.

**Strengths:**

1. The paper targets at an important problem of distillation of SR diffusion models. While diffusion distillation is a popular research area, it is interesting to see some insight particularly designed for SR models

2. The paper introduces a novel technique to reduce the bias of the score estimate of generated samples in SDS, which particularly fits in the insights from SR.

3. Empirical results shows promising improvements.

**Weaknesses:**

1. The biggest concern is insufficient baselines. The method compare against a large number of non-diffusion based methods or diffusion based iterative methods, but it lacks comparisons against the most closely related methods: other diffusion distillation algorithms. This method distill a pre-trained SR diffusion model into one step with some specific design for SR, but there are many distillation methods designed for general diffusion models, such as consistency model and the family of distribution matching distillation. The authors should run controlled experiment with the same teacher model with different algorithms to emphasize the relative advantage. For example, personally I found CM works well in distilling SR model into one step, and DMD and its variant can distilled the more complicated T2I model into one step. Their relative performance on SR diffusion is what we really care.

2. It seems like the method requires teacher model to generate clean samples, which can be computationally expensive, even if you pre-compute the data off-line.

3. The background of SDS and how to reduce the bias is unclear to readers without prior knowledge.

**Questions:**

N/A

---

> ### Author Response · Authors · 2024-11-20
> **Response to Reviewer eiDx**
>
> Thank you for your comments and feedback. We address your concerns here.
>
> >**Q1**: The biggest concern is insufficient baselines. The method compare against a large number of non-diffusion based methods or diffusion based iterative methods, but it lacks comparisons against the most closely related methods: other diffusion distillation algorithms. This method distill a pre-trained SR diffusion model into one step with some specific design for SR, but there are many distillation methods designed for general diffusion models, such as consistency model and the family of distribution matching distillation. The authors should run controlled experiment with the same teacher model with different algorithms to emphasize the relative advantage. For example, personally I found CM works well in distilling SR model into one step, and DMD and its variant can distilled the more complicated T2I model into one step. Their relative performance on SR diffusion is what we really care.
>
> >**A1**:
> Thank you for your valuable suggestion. We apply both consistency models and distribution matching distillation (DMD) to SR tasks for evaluation. Specifically, we employ consistency distillation under L2 loss and set the same boundary conditions as consistency models: $c_{skip}(t) = \frac{\sigma_{data}^2} {(\eta_t-\eta_0)^2 + \sigma_{data}^2}, c_{out}(t) = \frac{\sigma_{data}(\eta_t - \eta_0)}{\sqrt{\sigma_{data}^2+\eta_t^2}},$ which clearly satisfies $c_{skip}(0) = 1$ and $c_{out}(0) = 0$. For DMD, we alternately update the fake score network and generator, with the weights of the distribution matching distillation loss and regression loss set to 1.
>
> > The experimental results are presented in Table 1. As shown in the table, the high-resolution images generated by the model using consistency distillation are significantly inferior to those produced by other super-resolution methods across all metrics, which appears to contradict the reviewer's findings. We speculate that this discrepancy may be due to ResShift modifying the standard Markov chain of the diffusion model, making it difficult to apply consistency distillation directly. While applying DMD to super-resolution tasks has yielded promising results, it still falls short of our method. To further validate the effectiveness of our approach, we also transferred it to an unconditional generation task. The results of this evaluation on CIFAR-10 are presented in Table 2. As shown, our method achieves competitive performance, even in unconditional generation tasks, outperforming both consistency models and DMD.
>
> Table 1: Quantitative results of different SR methods. The best and second best results are highlighted in bold and italic. ∗ indicates that the result was obtained by replicating the method in the paper.
> |   Datasets  |   |  ImageNet-test |    |  RealSR  |    RealSR  |  RealSet65  |    RealSet65  |
> |-----------|:---------:|:---------:|:----------:|:---------:|:----------:|:---------:|:----------:|
> |   Methods   |  LPIPS  $\downarrow$   |  CLIPIQA $\uparrow$ |    MUSIQ $\uparrow$  |  CLIPIQA $\uparrow$   |    MUSIQ $\uparrow$   |  CLIPIQA $\uparrow$  |    MUSIQ  $\uparrow$  |
> |    LDM-15   |   0.269     |   0.512   |   46.419   |   0.384   |   49.317   |   0.427   |   47.488   |
> | ResShift-15 |   0.231     |   0.592   |    53.660  |  0.596   |   59.873   |   0.654   |   61.330   |
> |   SinSR-1   | **0.221**   |   0.611   |   53.357   | 0.689   |   61.582   |   0.715   |   62.169   |
> |   SinSR*-1  |   0.231     |   0.599   |   52.462   |  0.691   |   60.865   |   0.712   |   62.575   |
> |    DMD*-1   |   0.246     |  _0.612_  |   54.124   | _0.709_  |   _63.610_  |  _0.723_  |  _66.177_  |
> |   CD-L2*-1  |   0.568     |   0.192   |   27.002   |  0.230   |   30.578   |   0.262   |   35.101   |
> |   TAD-SR-1  |  _0.227_    | **0.652** | **57.533** |  **0.741** | **65.701** | **0.734** | **67.500** |
>
>
> Table 2: Generative performance on unconditional CIFAR-10. The best and second best results are highlighted in bold and italic.
> | Method | DDPM | DDIM | EDM(Teacher) | DPM-solver2 | UniPC | CD-L2 | CD-LPIPS |  DEQ |  DMD |  Ours  |
> |:------:|:----:|:----:|:------------:|:-----------:|:-----:|:-----:|:--------:|:----:|:----:|:------:|
> |   NFE $\downarrow$ | 1000 |  50  |      35      |      12     |   8   |   1   |     1    |   1  |   1  |    1   |
> |   FID $\downarrow$ | 3.17 | 4.67 |   **1.88**   |     5.28    |  5.10  |  7.90  |   3.55   | 6.91 | 3.77 | _2.31_ |

---

> ### Author Response · Authors · 2024-11-20
> **Response to Reviewer eiDx**
>
> >**Q2**: It seems like the method requires teacher model to generate clean samples, which can be computationally expensive, even if you pre-compute the data off-line.
>
> >**A2**: The generation of samples by teacher models does incur additional computational costs; however, these costs remain within an acceptable range, particularly when generating samples offline. We compare our method with SinSR in terms of training time. As shown in Table 3, when generating clean samples online, our training time is only two hours longer than that of SinSR, and the distillation process can be completed within a day. Moreover, generating samples offline further reduces both training time and computational resource consumption. Additionally, we compare the GPU memory usage of our method during training between offline generation clean samples and online generation clean samples. The results show that the online generation of clean samples increases GPU memory usage by less than 5%, which is within an acceptable range. Furthermore, because SinSR requires learning a bidirectional mapping between noise and images, its GPU memory usage is higher than that of our method.
>
> Table 3: A comparison of the training cost on 8 NVIDIA V100.
> | Method     | Num of Iters    | s/Iter | Training Time  |GPU memory (GB)
> |------------|:---------:|:----------:|:---------:|:----------:|
> | SinSR      |  30k  | 2.57  | ~21 hours |  17.30 |
> | Ours (Online)       | 30k   | 2.79  | ~23 hours | 11.72|
> | Ours (Offline)       | 30k   | 1.05  | ~9 hours | 11.17 |
>
>
>
> >**Q3**: The background of SDS and how to reduce the bias is unclear to readers without prior knowledge.
>
> >**A3**: Thank you for your valuable suggestion. In the revised manuscript, we will include more background information on SDS and provide a clearer explanation of how we address deviations in SDS.

---

> > ### Author Response · Authors · 2024-12-03
> > **Official Comment by Authors**
> >
> > Dear Reviewer eiDx:
> >
> > The discussion period between the authors and the reviewer is nearing its end, and we kindly request that you review our clarifications and revisions. If our response addresses your concerns, we hope you can reconsider your score.
> >
> > Thank you once again for your time and consideration.
> >
> > Best Wishes!
> >
> > Authors of Submission 1713

---

### Author Response · Authors · 2024-11-20
**Summary Response**

We thank all reviewers for their questions and constructive feedback. Based on these suggestions, we have made significant revisions to the manuscript.  Key changes in the revised submission include:

1. We have applied  DMD to super-resolution tasks and compared it with our method. The results are shown in Tables 2 and 3. (**Reviewer eiDX**, **Reviewer Rnto**)

2. We have included a more detailed explanation of the background knowledge related to score distillation sampling (SDS) technology in Section 2. (**Reviewer eiDX**)

3. We have incorporated PSNR and SSIM metrics for evaluation in the main experiments included in the revised manuscript. (**Reviewer gXos**, **Reviewer B832**)

4. We have conducted ablation experiments on the multi-scale features utilized by the discriminator, with the results presented in Table 9. (**Reviewer gXos**)

5. In addition to applying our method to distill the diffusion-based SR model ResShift trained from scratch, we also distilled the SD-based SR model SeeSR and compared it with other SD-based methods, such as OSEDiff. The results are shown in Tables 11, 12, and 13. (**Reviewer uBAa**, **Reviewer Rnto**)

6. We visualized the frequency spectra of the reconstruction results obtained by different methods through the Fourier transform to highlight the advantage of our method in generating high-frequency details. The results are presented in Figure 10. (**Reviewer uBAa**)

7. We have compared the inference time of TAD-SR distillation across different super-resolution models with their respective baseline methods, and the results are presented in Tables 6 and 14. (**Reviewer uBAa**, **Reviewer Rnto**)

8. We have provided more qualitative comparisons that contain fine details or small textures in Figures 9 and 15. (**Reviewer uBAa**)

9. We have carefully revised the motivation and methodology sections of the paper to enhance readability and clarity. Furthermore, we remain committed to ongoing revisions of our manuscript to enhance its readability and comprehensibility.(**Reviewer B832**, **Reviewer Rnto**)

10. We have provided the training process of the TAD-SR algorithm in the appendix to enhance the clarity of our method. (**Reviewer B832**, **Reviewer Rnto**)

11. We utilized samplers such as UniPC and DpmSolver to accelerate the teacher model and compare them with our method. The experimental results are presented in Tables 11, 12, and 13. (**Reviewer Rnto**)

12. We have included a discussion in the paper on the limitations of our proposed method and potential directions for future research. (**Reviewer Rnto**)

We hope that these changes strengthen the state of our submission.

---

> ### Author Response · Authors · 2024-11-27
> **Official Comment by Authors**
>
> We sincerely appreciate your valuable feedback and insightful suggestions, which have greatly helped us improve our manuscript. We have carefully addressed your concerns in our response and revised the manuscript accordingly.
> ﻿
>
> We understand that you have a busy schedule, but we would be grateful for any additional feedback or response you may have regarding our paper, as reviewer input is crucial for improving the quality and clarity of our work. Alternatively, if our revisions adequately address the issues raised, we kindly request a reconsideration of the score based on the clarifications and improvements made.
> ﻿
>
> Thank you once again for your time and consideration.
>
> Best Wishes!
>
> Authors of Submission 1713

---

### Meta-Review · Area_Chair_z4u1 · 2024-12-20

**Metareview:**

This paper receives mixed ratings of (5, 5, 5, 6, 6). The reviewers generally agree that the area this paper is exploring is interesting and meaningful, and the simplicity of the method, while having concerns about the comparison and improvement over existing works. The AC carefully read the paper, reviews, and rebuttal, and agree with the reviewers overall. In particular, in the response of the authors, the improvement over OSEDiff cannot be regarded as significant given a slower speed. As a result, the effectiveness of the methods could not be fully verified. While the AC agrees that this paper is an interesting exploration, the AC regretfully recommends a rejection.

**Additional Comments On Reviewer Discussion:**

Reviewers raise concerns mainly on the comparison and improvements, and the authors are managed to resolve most of the concerns. After reading the paper, review, and rebuttal, the AC feels that effectiveness of the proposed method cannot be convincingly verified, hence recommending a rejection.

---

### Decision · Program_Chairs · 2025-01-22

Reject